
**Overview of the first HyMeX Special Observation Period over Croatia**
*Branka Ivančan-Picek, Martina Tudor, Kristian Horvath, Antonio Stanešić, Stjepan Ivatek-Šahdan*
Meteorological and Hydrological Service
Grič 3, 1000 Zagreb, Croatia
**Abstract**
The HYdrological cycle in the Mediterranean EXperiment (HyMeX) is intended to improve the
capabilities to predict high impact weather events. In its framework, the first Special Observation
Period (SOP1), 5 September to 6 November 2012, was aimed to study heavy precipitation events
and flash floods. Here we present high impact weather events over Croatia that occurred during
SOP1. A particular attention is given to eight Intense Observation Periods (IOP)s during which high
precipitation occurred over the eastern Adriatic and Dinaric Alps. During the entire SOP1, the
operational models forecasts generally represented well medium intensity precipitation, while heavy
precipitation was frequently underestimated by the ALADIN 8 km and overestimated at higher
resolution (2 km). During IOP2 intensive rainfall event occurred in wider area of the city of Rijeka
in the northern Adriatic. Short-range maximum rainfall totals have achieved maximum values ever
recorded at Rijeka station since the beginning of measurements in 1958. The rainfall amount
measured in intervals of 20, 30 and 40 minutes could be expected once in a more than thousand,
few hundreds and hundred years respectively, and they belong to the extraordinarily rare events.
The operational precipitation forecast using ALADIN model at 8 km grid spacing underestimated
the rainfall intensity. Evaluation of numerical sensitivity experiments suggested that forecast was
slightly enhanced by improving the initial conditions through variational data assimilation. The
operational non-hydrostatic run at 2 km grid spacing using configuration with ALARO physics
package further improved the forecast. This article highlights the need for an intensive observation
period in the future over the Adriatic region, to validate the simulated mechanisms and improve
numerical weather prediction via data assimilation and model improvements in description of
microphysics and air-sea interaction.
**Keywords:** HyMeX SOP1, Adriatic TA, heavy precipitation, ALADIN mesoscale model, data
assimilation





## 1. Introduction

Special Observing Period 1 (SOP1) of the *HYdrological cycle in the Mediterranean Experiment –*
*HyMeX* project was performed from 5 September to 6 November 2012 (Drobinski et al., 2014). The
main objective of SOP1 was improving understanding and forecasting of the processes leading to
heavy rainfall and floods (Ducrocq et al., 2014). The characteristics of the Mediterranean region, a
nearly closed basin surrounded by highly urbanized and complex terrain close to the coast, makes
Mediterranean area prone to natural hazards related to the water cycle, including heavy precipitation
and flash-flooding occurring mostly in the late summer and autumn. Daily precipitation amounts
above 200 mm have been recorded during this season (e.g. Romero et al. 2000; Buzzi and Foschini
2000; Jansa et al. 2001, Ducrocq et al 2008). Within small and densely urbanized areas, intensive
and stationary precipitation events can rapidly result in dangerous floods, sometimes leading to
disastrous consequences (e.g. Silvestro et al., 2012; Rebora et al. 2013; Ivančan-Picek et al. 2014).
This stresses the importance of such events through their impact on social and economic
circumstances of local communities. Numerical weather prediction (NWP) models have made a
significant progress through the development of convection permitting systems. However, the
ability to predict such high-impact events remains limited because of the contribution of fine-scale
processes not represented in NWP models, and their interactions with the large-scale processes, as
well as limitations of the data assimilation and especially for the convective-scale data assimilation.
HyMeX aims to improve our understanding of precipitating systems, especially processes
responsible to their formation and maintenance, as well as to improve the ability of numerical
weather prediction models in forecasting the location and intensity of heavy precipitation events in
the Mediterranean.

The orography and thermal contrast of the Mediterranean basin together with approaching upper-
level trough frequently induce lee cyclogenesis (e.g. Buzzi and Tibaldi, 1978; Horvath et al., 2006)
and provide a trigger mechanism for a range of extreme weather phenomena, such as local
downslope Bora windstorms (known as Bura in Croatia) (e.g. Grisogono and Belušić, 2009), strong
winds Scirocco and Tramontana (Jurčec et al. 1996; Pandžić and Likso 2005; Jeromel et al., 2009)
orographic precipitation, thunderstorms, supercells and mesoscale convective systems (Ivančan-
Picek et al. 2003; Mastrangelo et al., 2011), and water-spouts (Renko et al., 2012). Heavy
precipitation occurs preferentially downstream of a cyclone aloft (Doswell et al., 1998).

The seasonal distribution of heavy precipitation suggests the relevant role of the high sea surface
temperature (SST) of the Mediterranean Sea during the autumn season, when the lower layer of the



atmosphere is loaded with water vapour. The large thermal gradient between the atmosphere and the
sea favours intense heat and moisture fluxes, which are the energy source for storms (Duffourg and
Ducrocq, 2013). As the sea provides a large source of moisture and heat, the steep slopes of the
surrounding mountains in the vicinity of highly urbanized coastal areas of the Mediterranean are the
key factors in determining the moisture convergence and the rapid uplift of moist and unstable air
responsible for triggering condensation and convective instability processes (e.g. Rotunno and
Ferretti, 2001; Davolio et al., 2009). The coastal mountains, however, are not the only sources of
lifting. Favourable synoptic upper-level setting, frontal lifting associated with quasi-stationary
frontal systems and lower tropospheric mesoscale convective lines may also induce the convective
instability.

One of the key components of HyMeX is the experimental activity, which is aimed at better
quantification and understanding of the water cycle in the Mediterranean with emphasis on intense
events. Over the whole Mediterranean region three target areas (TA) have been proposed for
Enhanced Observational Period (EOP) to provide detailed and specific observations for studying
key processes of the water cycle (http://www.hymex.org). One of them is the Adriatic Sea and
Dinaric Alps (Adriatic TA) which has been proposed for the study of heavy precipitation events and
flash-floods and considerable effort from the Croatian meteorological community was put into the
campaign.
The Adriatic Sea is a northwest–southeast elongated basin in the central Mediterranean sea,
approximately 200 km wide and 1,200 km long and is almost entirely enclosed by mountains,
namely the Apennines to the west and southwest, the Alps to the north and the Dinaric Alps to the
east and southeast. These topographic features play a large role in the structure and evolution of the
weather systems associated with heavy precipitation (e.g. Vrhovec et al., 2001; Ivančan-Picek et al.
2014). This area is one of the rainiest in Europe with expected annual amounts of precipitation
greater then 5.000 mm in the mountainous hinterland on the south (end) part of the Adriatic Sea
(Magaš, 2002).

Although the Adriatic TA was not a part of extensive experimental activity during the SOP1, many
events that affected the Western Mediterranean expanded at the Adriatic area too. During SOP1, 16
IOPs were dedicated to heavy precipitation events (HPE) over France, Spain and Italy and many of
these events subsequently affected the eastern Adriatic Sea and Croatia.

The aim of the paper is: 1. to provide a scientific overview of the HPEs that affected the Adriatic TA



during SOP1; 2. to provide and examine the operational numerical models skill of the precipitation
forecasts in Croatia; 3. to provide a detailed description of the extraordinarily rare heavy
precipitation event IOP2.

The remainder of this paper is organised as follows. Section 2 describes the area of Dinaric Alps
and the Adriatic region, measured and model data provided by Croatian Meteorological and
Hydrological Service (DHMZ). Section 3 analyses the events during HyMeX SOP1 that produced
more than 100 mm of precipitation during 24 hours on eastern Adriatic coastline. Performance of
the operational precipitation forecasts is assessed through verification of forecasts mostly with the
Croatian surface observation network. In Section 4 an additional attention is given to the
extraordinarily rare heavy precipitation event IOP2.
Finally, we analyse and discuss the potentials for improving numerical weather predictions through
data assimilation using sensitivity experiments. The summary and conclusions are reported in
Section 5.

**2. HyMeX SOP1 in Croatia: observations and models**

Mediterranean is one of the climatically most pleasant areas in the world. Nevertheless, the area is
prone to high-impact weather phenomena, affecting people´s lives and activities and causing
extensive material damage. This context was favourable for an active participation of the Croatian
scientific community in the HyMeX project. Croatian research community was active in the
preparation of the scientific programme included the identification of typical weather patterns over
the regions and the target areas. During the SOP1, the national meteorological service supported the
main HyMeX Operational Centre (HOC) in Montpellier (France), through visiting scientists and
their meteorological expertise as well as providing observations, numerical modelling products and
forecast data.

This section summarizes the observational network in Croatia operational during SOP1 and the
operational forecasting modelling chain producing numerical weather predictions during SOP1.


**2.1.   Observations**

The instrumentation deployed over the Adriatic TA during the SOP1 belongs mainly to the
observational network of DHMZ. DHMZ deployed a ground observation operational network that





includes automatic, climatological and raingauge stations, two radio-soundings (Zagreb-Maksimir
(station ID = 14240, H = 123 m asl, $\varphi = 45^0 49'$N, $\lambda = 16^0 02'$E) and Zadar-Zemunik (station ID =
14430, H = 88 m asl, $\varphi = 44^0 5'$N, $\lambda = 15^0 21'$E)) and two radars (Bilogora and Osijek).

The meteorological measurements and observations on 58 SYNOP stations are done every hour and
reported in real time during the SOP1. Majority of SYNOP stations are also equipped with an
automatic station. All automatic stations measure data with a 10 minute interval and report the
measured data in real time. However, not all 63 automatic stations measure all the meteorological
parameters. There are 21 automatic stations that report only the wind parameters (average 10
minute speed and direction, wind gust speed measured in the last 10 minutes). Five more stations
measure the wind parameters, temperature and relative humidity. All real time surface measurement
(SYNOP, and automatic station data), and available radar figures are stored in HyMeX data centre.

The dense network of climatological stations is the source of temperature, humidity and wind speed,
cloudiness and visibility are estimated by observation only 3 times a day at 0600, 1300 and 2000
UTC; accumulated rainfall and snow height are measured at 0600 UTC (there were more than 500
stations reporting accumulated 24-hourly rainfall).

In addition to operational radio-sounding in Zadar-Zemunik  at 0000 and 1200 UTC several extra
radiosoundings were deployed through Data Targeting System (DTS) upon request of the HOC.
These targeted radiosoundings, among others in the western Mediterranean, were activated during
IOP16, which caused heavy precipitation, strong winds and snow in the eastern Adriatic. The
requests for additional radiosoundings at 0600 and 1800 UTC were carried out under the
EUMETNET Observation Programme. Sounding data measured at Zadar-Zemunik, located on the
eastern coast of the Adriatic Sea at the southern end of Velebit Mountain, provided information on
the vertical structure of the troposphere in order to monitor the upstream flow of the precipitation
events in the Adriatic region. The selection of sensitive area predictions (SAP) used methods
developed by ECMWF and Meteo-France (Prates et al., 2009). The verification area selected for
SAP calculations was centred over the northern and/or central Adriatic.

To complement the ground-based observations, the data from two radars in Croatia (Bilogora
(H=270 m asl, $\varphi = 44^0 53'$N, $\lambda = 17^0 12'$E) and Osijek (H=89 m asl, $\varphi = 45^0 30'$N, $\lambda = 18^0 34'$E)) and
one in Slovenia (Lisca; H=944 m asl, $\varphi = 46^0 04'$N, $\lambda = 15^0 17'$E) are available operationally in a
graphic form. The estimation of the instantaneous surface rain rate from Lisca and Bilogora radars
were provided to the HyMeX web server in real time. Northwest Croatia, particularly Rijeka and





Istria are covered by operational radars in Croatia, Slovenia and Italy but the area is on the edge of
the ranges and behind a mountain obstacle.

The standard Meteosat Second Generation (MSG) Spinning Enhanced Visible and Infrared Imager
(SEVIRI) data are available with an interval of 15 minutes and Rapid Scan Service (RSS) data are
available with 5 minute interval. The abundance of remote sensing data on the HyMeX server
encourages detailed analyses of all the cases that produced HPEs over Croatia during SOP1.

Satellite derived precipitation data are used as provided from the Tropical Rainfall Measuring
Mission (TRMM, Huffman et al., 2007). In particular we used the 3 hourly accumulated
precipitation data from the 3B42RT product to compute the 24 hourly accumulated rainfalls for the
period from 0600 UTC to 0600 UTC the next day, and 1 hourly precipitation data from 3B41RT
product to compare it with the precipitation forecast by operational numerical weather prediction
models.
**2.3  Mesoscale models**

During the SOP1, DHMZ provided the products from the operational forecast (Tudor et al., 2013).
At the time it consisted of ALADIN model (Aladin International Team, 1997; Tudor et al. 2013) run
twice per day on a domain in 8 km resolution (Figure 1a) starting from 0000 and 1200 UTC
analyses up to 72 hours lead time. The operational suite used lateral boundary conditions from the
global model ARPEGE run operationally in Meteo-France. The initial fields are obtained using data
assimilation procedure (Stanešić, 2011). The high 2 km resolution forecast (Tudor and Ivatek-
Šahdan, 2010) using ALADIN model with non-hydrostatic dynamics (Benard et al 2010) with the
physics package that included the convection scheme was running operationally during the HyMeX
SOP1 campaign (Figure 1b). The convection scheme used in the high-resolution model is modular
multiscale misrophysics and transport (3MT) scheme for precipitation and clouds (Gerard and
Geleyn, 2005; Gerard, 2007; Gerard et al., 2009). Both runs use SST from the initial file of the
global model ARPEGE forecast.

A short description of the models characteristics and the operational set-up during SOP1 is given in
the following sub-sections.






### 2.3.1 Operational 8 km ALADIN forecast

The operational ALADIN model is a limited-area model that applies Fourier spectral representation of the model variables using fast Fourier transforms (FFTs) in both directions with a quadratic elliptic truncation (Machenhauer and Haugen, 1987) that ensures an isotropic horizontal resolution and that the nonlinear terms of the model equations are computed without aliasing. The forecast in 8 km resolution is run on a domain with 240x216 grid points that includes a band of 11 points along northern and eastern boundaries with unphysical terrain created for the biperiodization (Figure 1a). The dynamical computations are performed using semi-implicit semi-lagrangian discretisation (Robert, 1982) to solve the hydrostatic dynamics and finite difference method on 37 levels of hybrid pressure type eta coordinate (Simmons and Burridge, 1981) in the vertical. The operational physics package at the time used prognostic TKE, cloud water and ice, rain and snow and diagnostic scheme for deep convection. The prognostic equations for condensates are solved using the barrycentric approach (Catry et al., 2007).

A physical horizontal diffusion scheme, called semi-Lagrangian horizontal diffusion (SLHD), is based on the physical properties of the flow (Váňa et al., 2008) consists of combining two semi-Lagrangian interpolators of different diffusivity with the flow deformation as a weighting factor. The primitive prognostic equations are solved for the prognostic variables using the stable extrapolation two time level, semi-implicit, semi-Lagrangian advection scheme (SETTLS, Hortal 2002) with a second-order accurate treatment of the nonlinear residual (Gospodinov et al., 2001).

The coupling of the model variables along the lateral boundaries is done using relaxation scheme (Davies, 1976) in a zone 8 grid points wide using time dependent and periodic LBCs (Haugen and Machenhauer, 1993) at the end of the grid-point computations (Radnoti, 1995) due to constraints imposed by the model dynamics. The prognostic LBCs were operationally taken from the global model ARPEGE (while there were alternative LBCs available from IFS too). The initial conditions are computed using 3dVar for the upper air fields (Hollingsworth et al 1998; Lorenc, 1986) and optimal interpolation for surface. The operational background error matrix for 3dVar data assimilation for the ALADIN weather prediction system at DHMZ was calculated using standard NMC procedure (Parrish and Derber, 1992; Bölöni and Horvath, 2010). Observations used in the data assimilation system include ground station observations (2 metre temperature, 2 metre relative humidity, pressure), radiosoundings (temperature, humidity, wind components), aircraft-based observations (temperature, wind components), wind components derived from a cloud motion detection process based on the measurements of geostationary satellites and brightness temperature



coming from geostationary and polar satellites. Around ~15000 observational data per assimilation
cycle remains active after observation pre-processing.

In the physics parameterization package used operationally, there is a simple microphysics scheme
with prognostic cloud water and ice, rain and snow (Catry et al., 2007) combined with a statistical
approach for sedimentation of precipitation (Geleyn et al., 2008). The operational radiation scheme
(Ritter and Geleyn 1992) based on Geleyn and Hollingsworth (1979) and enhanced recently
(Geleyn et. al. 2005a, 2005b) is simple and computationally cheap using only one spectral band for
computation of the long-wave and one for short-wave radiation. The turbulence contribution is
computed using prognostic TKE (turbulent kinetic energy) according to Geleyn et al. (2006),
modified from Louis et al. (1982) type scheme and includes a contribution of the shallow
convection (Geleyn, 1987). The exchange with surface is computed using the Interaction Soil
Biosphere Atmosphere (ISBA) surface scheme (Noilhan and Planton, 1989) that is also used in the
surface data assimilation (Giard and Bazile, 2000). Wind, temperature and humidity on the heights
of the standard meteorological measurement (10 and 2 meters above ground) are computed by
interpolation from the lowest model level (about 17 meters above ground) using a parameterised
vertical profile (Geleyn, 1988) dependent on stability.

**2.3.2 Operational 2 km non-hydrostatic ALADIN forecast**

Upon numerous case studies of severe weather events (e.g. Tudor and Ivatek-Šahdan, 2010),
additional operational forecast run was established in July 2011 that uses ALADIN with non-
hydrostatic dynamics and a complete set of physics parameterisations, including the convection
scheme. Most of the setup in dynamics and physics is similar to the 8 km resolution operational
forecast, apart from the non-hydrostatic dynamics (Benard et al., 2010). The 8 km resolution run
used operationally a diagnostic convection scheme (Geleyn et al., 1995), while the 2 km resolution
run used a prognostic convection scheme (Gerard and Geleyn, 2005; Gerard, 2007) that allows
combining resolved and convective contributions in the grey zone (Gerard et al., 2009). The
convective scheme available for the operational forecast in 2012 did not use the complete
prognostic convection scheme with prognostic entrainment, but only prognostic mesh fractions and
vertical velocities in updraft and downdraft. This forecast run is computed once per day, following
the 0000 UTC operational 8 km resolution forecast. It uses the 6 hour forecast from the 8 km
resolution operational run as input initial file and is initialized using scale selective digital filter
initialization (SSDFI, Termonia 2008). Hourly 8 km forecast interpolated to 2 km resolution are
used as LBC (lateral boundary conditions) files. The forecast range is 24 hours, until 0600 UTC on



the next day that allows comparison to precipitation data from the rain-gauges available in the high
resolution network. Taking into account limitation due to computer resources and time constraints
of forecast availability this setup was made in order to provide as soon as possible high resolution
forecast for current day. Future upgrades will include longer time range of forecast and initialization
by implementing data assimilation procedure. While other approaches such as data assimilation
cycle at 2 km grid spacing, or initiation of the 2 km model with 8 km analysis at 0006 UTC may be
more favourable, the described set-up is designed to mitigate the insufficient computing resources
available.


## 3.  Heavy precipitation events over the Adriatic TA during SOP1



In late summer and early autumn 2012 (from 5 September to 6 November), Hymex SOP1 which
was dedicated to heavy precipitation and flash floods took place over the western Mediterranean
(Ducrocq et al, 2014). During SOP1 20 IOPs were declared and 8 of these events affected the
Adriatic TA (Table 1). Most of these events (6 IOPs) were dedicated to HPEs over the northern
Adriatic (city of Rijeka).
Figure 2a shows the total precipitation measured by the Croatian rain gauge network cumulated
over the whole SOP1. The total precipitation for the SOP1 was above the corresponding
climatology (Zaninović et al., 2008) for September and October for Adriatic TA. Similar was found
over the Apennine peninsula (Davolio et al., 2015). Maximum of precipitation during SOP1 was
recorded on the northern Adriatic (city of Rijeka) and its mountainous hinterland of Gorski Kotar
(more than 1000 mm at some locations). There were 15 days with daily rainfall accumulations
exceeding 100 mm at locations in the Adriatic TA (Figure 2b). There were more IOPs dedicated to
HPEs over the Adriatic TA in October than in September 2012 which was also the case in the
western Mediterranean (Ducrocq et al., 2014). Several of these events caused local urban flooding
(Rijeka, Pula and Zadar) with considerable material damage.
Some of the IOPs were embedded in a synoptic setting characterized with cyclones over the western
Europe and Mediterranean recognized as a favourable conditions to heavy rainfall (e.g. Dayan et al.
2015). The storm tracks of these cyclones coming from the North Atlantic to Europe depend on the
direction and strength of the westerly winds controlled by the relative positions of the permanent
Azores High and Icelandic Low. The variation of this relative position is known as the North
Atlantic Oscillation (NAO) index, taken to be positive when the difference in pressures measured at
Azores and Iceland is high and related to strong westerly wind. NAO is negative when the pressure
difference is low, westerlies are reduced and storm tracks are shifted towards the Mediterranean





increasing the frequency and intensity of rainfall there. During the first weeks of the SOP1, NAO
was in a slightly positive phase and few HPEs happened before 22 September affecting only the
eastern part of the basin, particularly Adriatic area and Dinaric Alps as well as the Italian target
areas. After that period, the weather above North Atlantic was characterized by mostly negative
NAO and long lasting hurricane Nadine (16 September to 3 October 2012). Hurricane Nadine
affected the weather not only over the Azores region it traversed, but it also modified the Rossby
wave breaking downstream (Pantillon et al. 2015) and possibly reduced the long term predictability
over Mediterranean.
**3.1 Overview of IOPs over the Adriatic TA**
The influence of different meteorological characteristics and physical processes that produced HPE
over Adriatic target area and Dinaric Alps are briefly analysed and summarized. Previous research
on HPE occurrence in the wider Adriatic region (e.g. Doswell et al., 1998; Romero et a., 1998;
Vrhovec et al., 2001; Kozarić and Ivančan-Picek, 2006; Horvath et al., 2006; Mastrangelo et al.,
2011; Mikuš et al., 2012) highlighted *cyclonic activity* in the western Mediterranean and in the
Adriatic as a triggering mechanism for a range of extreme weather phenomena including HPE.
Position of cyclones that appear in the Adriatic Sea basin strongly influence the climate and weather
conditions in the area (Horvath et al., 2008).
During the SOP1 several troughs entered the western Mediterranean and produced cyclogenesis
over the Gulf of Genoa or over the Tyrrhenian Sea (IOP2, IOP4, IOP16, IOP18, see Figure 3).
Among those events, IOP16 and IOP18 represent excellent cases for the science issues identified in
HyMeX program for the western Mediterranean (convection initiation, cloud-precipitation
processes, air-sea coupled processes). These situations produce favourable conditions for HPE on
the southern side of the Alpine ridge including the northern Adriatic region.
During these events, the Adriatic TA was strongly affected by the Genoa cyclone. Figures 3d and 3e
show the sea level pressure and low-level wind vectors at 1200 UTC on 27 and 31 October. The
low-level wind field was dominated by a low-level jet stream that carried the warm and humid
Mediterranean air to the Adriatic Sea. This situation was favourable for the strong S-SE sirocco
wind (IOP18) known as the *jugo* in Croatian (e.g. Jurčec et al., 1996).
During IOP16, targeted radio-soundings aimed at both data assimilation, case analysis and
verification were deployed over the central Mediterranean area and Adriatic area. The time
evolution of the vertical structure of troposphere on the eastern Adriatic coast is inferred by DTS
deployed and standard radiosoundings at Zadar-Zemunik during 26-28 October (Figure 5). Gradual



moistening of the lower troposphere occurred during 26 October during southeasterly near-surface
*jugo* wind in the Adriatic basin and southwesterly flow aloft. The air column below 500 hPa was
nearly saturated and also rather moist above. On 26 October this moistening was still not associated
with significant values of convective available potential energy (CAPE). On the next day, however,
CAPE increased to over 1200 J/kg on 1200 UTC and over 1000 J/kg on 1800 UTC 27 October. The
winds strengthened throughout the troposphere, and the highest intensity was observed in the layer
between 300 and 200 hPa. Strong southwesterly shear of approximately 20m/s in the first 2 km of
the troposphere was also present over this area.

Both IOPs (IOP16, IOP18) were fairly well forecast (Figure 4). The precipitation timing and the
location of the maxima are reproduced quite well in the models forecasts. In less than 24 h, intense
precipitation exceeding 170 mm affected the northern Adriatic area. Operational forecast of the 2
km model resolution run overestimates rainfall above mountains, but it is consequently closer to the
extreme amounts in the Rijeka area (Figure 4-B and Figure 4-C).

Occasionally a mesoscale cyclone associated with a potential vorticity (PV) anomaly formed over
the Adriatic Sea developed (IOP4). This event was enhanced by the Bora flow over the northern
Adriatic Sea and warm southerly wind on the southern Adriatic (Figure 3a). Mesoscale cyclone
moved slowly south eastward inducing instability over central Adriatic Sea, with intense convective
phenomena on both sides of the basin. Several rain gauges stations reached maxima of over 150 –
200 mm/24h along the eastern Italian coast (Maiello et al., 2014) and more than 100 mm/24h was
recorded over southeast coast of the Adriatic with the maximum over Pelješac peninsula. There are
also local maxima in precipitation located above the sea that can be recognized in the precipitation
estimates from the measured satellite data (Figure 6b).

Several events were characterized by frontal lifting associated with quasi-stationary ***frontal systems***
which help the release of convective instability (IOP9, IOP12, IOP13). This weather regime
provided a favourable environment for HPE with thunderstorms over the northern Adriatic Sea
where 127.4 mm/24h was recorded in the city of Rijeka in the northern Adriatic (IOP9). After 8
days without HPE in the first part of October, weather patterns became favourable to HPE over the
Adriatic TA with blocking and SW advection of warm and moist air. HPE occurred every day from
11 to 16 October (IOP12, IOP13). Smooth troughs entering the western Mediterranean Sea were
observed producing a south westerly flow over the Adriatic TA. A cold front was moving eastward
supporting the advection of moist air on the low levels towards coastline. This warm and moist air
ahead of the front supported intensive convective activity that formed a rain band stretching from





Tunisia over southern Italy to southeast Croatia and caused intensive precipitation with more than
100mm/24 h mostly above open sea and several outermost islands (Mali Lošinj (H = 53 m asl, φ =
$44^0 53'$N, λ = $14^0 48'$E), Silba (H = 20 m asl, φ = $44^0 37'$N, λ = $14^0 7'$E)).
Large-scale conditions such as found in these IOPs help to generate mesoscale and local processes
which modify additionally flow regimes leading to quite different precipitation patterns.

Several events were characterised by convection over the sea followed by ***orographic precipitation***
(IOP2, IOP9, IOP16, IOP19). During the whole IOP19 (3-5 November 2012) south westerly
advection of warm and humid air produced convection over the northern Adriatic and orographic
precipitation along the Kvarner bay. Detailed synoptic situation is described in Ferretti et al. (2014)
and Davolio et al. (2016). A south westerly flow over the whole region of the western
Mediterranean was produced by a baroclinic wave that formed over northwest Europe to northern
Africa due to weakened westerlies and low NAO. Strong southwest flow in lower troposphere
ahead of the cold front supported advection of moist and warm air. More rainfall was recorded on
rain gauges on the north eastern Adriatic coast. During this event 177.0 mm/24h was recorded in
Klana, the hinterland of the city of Rijeka (Figure 4-D) and the precipitation was mainly orographic
forced with strong southeast *jugo* (sirocco) wind (Figure 3f). Previous studies (e.g. Buzzi and
Foschini, 2000; Ivančan-Picek et al., 2014; Davolio et al., 2016) show that the largest component of
the mountain-range-scale precipitation appears to be due to the orographic lift of the moist and
impinging low-level flow. Consequently, the vertical uplifts forced by the Dinaric Alps area were
favourable for convection initiation and maintenance. Coastal mountains close to the Adriatic Sea
are not the only sources of lifting. The low-level circulation over the sea frequently generates low-
level convergence responsible for convective initiation (Jansa et al., 2001; Davolio et al. 2009).
Mesoscale cyclone or frontal system moved slowly south eastward inducing instability over central
Adriatic Sea usually is the cause of the strong low-level ***convergence*** between the southerly *jugo*
(sirocco) and northeasterly *bora* wind. This was the case during IOP4 when more than 100mm/24h
was recorded over the southeast Adriatic coast and above the open sea (Figure 6b).

The sirocco wind is the cause of a piling up of Adriatic water near the northernmost coasts that
occasionally floods the city of Venice (Orlić et al., 1994). This was the case also during the IOP16
and IOP18. The Venice Lagoon was hit by the "acqua alta" (high water), the warning level was
exceeded twice with more than 120 mm on 27 and 28 October (Ferretti et al., 2014) and more than
140 mm was measured on 1 November 2012. The cyclone during IOP16 caused the lowest pressure
recorded over the Adriatic TA during the whole SOP1 (Figure 3d). Advection of the warm air
combined with intensive advection of cyclonic vorticity contributed to the orographically induced





upward motion in the area of the northern Adriatic and the adjacent mountains, which resulted with
HPE in city of Rijeka and mountainous hinterland (180 mm/24h).

In IOP4 heat loss caused by strong *bora* wind was very intensive. *Bora* was severe on northern
Adriatic, exceeding 24 m/s. Strong *bora* wind brings cold and dry continental air over the warm
Adriatic basin, generating intense air-sea heat exchanges and sea surface cooling (e.g. Grisogono
and Belušić, 2009) during short time. The proper representation of sea surface temperature (SST) in
the numerical models, especially in small and shallow basins like Adriatic Sea is necessary for
improving the short-range precipitation forecasts (e.g. Davolio et al., 2015b; Stocchi and Davolio,
2016; Ricchi et al., 2016). The response of heavy precipitation to a SST change is complex and
mainly involves the modification of the boundary layer characteristics, flow dynamics and its
interaction with the orography. In the numerical modelling the SST representation is generally
unrealistic and usually keeps SST fixed at its initial value. Figure 6a shows SST measured on
station Bakar close to the city of Rijeka for the whole SOP period. During IOP4 (13 – 14 September
2012) the SST decreased for 10 °C on station Bakar in comparison to representation in operational
model which use LBC from the global model ARPEGE. Therefore, the SST near the coast was
colder than in the ALADIN model forecast affecting the ability of the forecast model to properly
forecast the meteorological fields there. In addition to operational SST, control simulation is driven
by SST field provided through the OSTIA analyses (Donlon et al., 2012). The daily accumulated
precipitation for operational 2 km model run and the control simulation with modified colder SST
from OSTIA are presented at Figure 6d and 6e.  In this case, control simulation run is more realistic
(see Figure 6b) and generally drier than the operational with a warmer SST. Colder SST resulted
with decreasing of precipitation over the mountainous Adriatic cost.
IOP4 shows the needs for further improvements of the role of SST and surface (latent and sensible)
heat fluxes over the Adriatic Sea, which attain large values during strong *bora* events. However a
more detailed analysis of the impact of SST on precipitation is ongoing.


**3.2. Verification of the precipitation forecasts during SOP1**

Performance of the operational precipitation forecasts with the ALADIN model at 8 km and
ALADIN model at 2km grid spacing during SOP1 was assessed by comparing forecasts with the
measurements from Croatian surface observation network. Model results were compared with 24-
hour accumulated precipitation measured by the rain gauges where before the calculation of
verification scores results for ALADIN 2 km was upscaled to ALADIN 8 km grid to avoid double
penalty. Contingency tables (Table 2 and 3) were evaluated with three categories defined according





to the amount of 24h accumulated precipitation and classified as dry, medium and strong. An event
was defined as dry if 24 h accumulated precipitations on the rain gauge station was less or equal 0.2
mm/24h. The border between the medium and strong categories was defined as the 95[th] percentile
of measured 24h accumulated precipitation (50.42 mm/24h) during the SOP1 period, but with dry
events excluded.
Figure 7 presents 24 hour accumulated precipitation histograms from both models and rain gauges
during the whole SOP1 period and during the specific days corresponding to 8 IOPs indicated in
Table 1. Measurements show that during the entire SOP1 period, large percentage of events was dry
(64.7%). Value corresponding to the 95[th] percentile (50.4 mm) is indicated at graph and it appears as
a reasonable threshold for heavy precipitation events that we want to verify. Histogram for IOP days
only (8 IOP cases) as expected show that the number of dry events is reduced (18.1%) and relative
frequency of events shifts towards events with higher amounts of precipitation.

While for the whole SOP1 period ALADIN 8 km model distribution is in rather good agreement
with rain gauge measurements except for most intensive rain, the model distribution for the IOP
days only shows that the model tends to underestimate frequency of week and strong precipitation
events while it overestimates frequency of moderate precipitation events. For ALADIN 2 km SOP1
and IOP days only histograms shows similar results where the model tends to underestimate
moderate precipitation while at the same time it tends to overestimate strong precipitation.
Comparison of two models shows a better agreement of ALADIN 2km model with measurements
especially for very week and strong precipitation.
In Table 2 and 3 verification measures (Wilks, 2006) calculated from comparison of 24 hour
accumulated precipitation from rain gauges and model, for the three categories and for different
periods are summarized. As most of measures are Base Rate (BR) sensitive and they can be safely
used only to compare two models for the same event, polychoric correlation coefficient (PCC; Juras
and Pasarić, 2006) as additional measure was calculated because PCC does not depend on BR or on
frequency bias (FBIAS). For both ALADIN models PCC show rather high level of association
between observations and forecast for whole SOP1 while it has smaller value for only IOP days.
For both models smallest value of PCC is for IOP 9 where both models overestimated number of
strong precipitation events, especially ALADIN 2 km which can be seen from much higher FBIAS
than the one from ALADIN 8 km model. Comparing performance of two ALADIN models, it can
be seen that ALADIN 2 km has higher level of association between observations and forecast for
IOP13 and IOP19 compared to ALADIN 8 km. For IOP13 ALADIN 2 km is relatively more
accurate in all three categories which can be seen from higher values of critical success index (CSI).
For IOP19 FBIAS values show that ALADIN 2 km overestimates frequency of strong precipitation



but at the same time it is relatively more accurate for other two categories (higher CSI). For dry
category ALADIN 2km has better scores for almost all selected cases (higher CSI; FBIAS closer to
1). For medium precipitation ALADIN 8 km has better scores except for IOP13 and IOP19. For
strong category scores show that ALADIN 2 km tends to overestimate frequency of strong events
while ALADIN 8 km tends to underestimate frequency of strong events with only exception for
IOP19 where both models overestimated number of strong precipitation events (especially
ALADIN 2 km).


## 4. IOP2 over north-eastern Adriatic TA




Although the Adriatic TA was not a part of extensive experimental activity during the SOP 1, many
events that affected the Western Mediterranean expanded at the Adriatic area too. During the IOP 2
in the late evening hours of September 12 a rainy episode with very heavy rain falling over only a
few hours have been recorded over the city of Rijeka at the northern cost of Kvarner Bay in the
eastern Adriatic sea and its mountainous hinterland of Gorski kotar. According to the report of the
Municipal Water and Sewer Company of the city of Rijeka  some major city roads became rivers
and streams, sewage manhole covers were discharged and massive caps flew into the air up to two
meters, and then a spate of them were carried up to one hundred meters away from the shaft.
Ferretti et al. (2014) described IOP2 in north eastern Italy (NEI) and analysed the meteorological
characteristics and synoptic situation. A shallow orographic cyclone developed in the lee side of the
Alps extending from the Genoa Gulf to the northern Adriatic. Simultaneously with the Genoa
cyclogenesis, a twin type of cyclone (Horvath et al., 2008) developed in the northern Adriatic
(Figure 8 a,b). The Croatian coast of northern and middle Adriatic was influenced by the strong
moist south-western flow on the leading side of the cyclone(s). The air was moist due to southwest
advection and evaporation from the Mediterranean. Below 2 km there was strong convergence over
the northern Adriatic. Due to its specific position deep in Kvarner bay which is open from the
southwest and, at the same time, in the very pedestal of the Velebit mountain chain, the city of
Rijeka and its surroundings have the geographic preconditions for pronounced convection with
extensive precipitation in such specific synoptic conditions (e.g. Ivančan-Picek et al., 2003).
During the day in the late afternoon cold air irrupted along the Alpine slopes together with the
passage of the cold front over NEI and north eastern Adriatic Sea resulted with intensive convective
processes.




### 4.1. Extreme value analysis of short-term precipitation maxima



Spatial distribution of daily rainfall amounts for the IOP2 rain episode indicates that the largest
amounts fall over the city of Rijeka (220 mm at the meteorological station Rijeka located 120 m
above sea level), and the surrounding mainland hilly slopes and mountainous hinterland. According
to the recorded rainfall data by ombrograph at the meteorological station Rijeka the more detailed
insight into the temporal rainfall distribution during the short-term interval of this heavy rainfall
event is possible (Figure 9). The rainfall episode that occurred during the six-hour period between 6
pm and midnight, experienced its most intense part between 9 pm and 11 pm. Maximum 20, 30, 40,
50, 60 and 120 minutes rainfall totals, which belong to this most intense part of the rainfall episode,
have not been recorded at Rijeka station since the beginning of measurements in 1958 (Table 4).
Especially intense were the rainfall intervals of 20, 30 and 40 minutes that could be expected once
in a more than thousand, few hundreds and hundred years respectively and they belong to the
extraordinarily rare event, computed from the period 1958 – 2011 (Patarčić et al., 2014). The
maximum amounts that fall in the interval of two and four hours could be expected ones in forty
and fifty years respectively.

### 4.2 Observational analysis



On 12ᵗʰ September 2012 a sequence of convective events hit the northeastern part of Italy and in
particular the eastern part of Veneto and the plain of Friuli Venezia Giulia regions. During the day at
least two events could be classified as supercells, the first one being also associated with a heavy
hail fall (Manzato et al., 2015). After a few hours, a third storm system, resembling a squall-line,
although of limited dimensions, swept over the area.
EUMETSAT was conducting its first experimental 2.5-minute rapid scan with the MSG-3 satellite,
with data available from early morning until 0900 UTC of the IOP2 day. Unfortunately, the
experimental rapid scan data with 2.5 minute interval taken by MSG-3 satellite (renamed to
Meteosat-10) were available only until 0900 UTC 12 September 2012.
Nearby area of Istria and Rijeka received the first rain in the early afternoon that soon stopped
before the torrential rain in the evening, between 2100 and 2300 UTC. The last one is connected to
the third storm over Italy (as discussed in Manzato et al. 2015) that was an elongated storm moving
along the coast of north Adriatic. Convection developed over the northern Adriatic and warm and
moist advection produced intensive precipitation triggered by the orography inland.

Satellite data show formation of cumulonimbus clouds (Figure 10). This intensive rainfall band


reached Trieste and Slovenia according to radar figures (not presented) and merged with the rainfall
band that formed above Trieste at 1800 UTC. Another rainfall band formed above Istria peninsula at
1930 UTC. Intensive rainfall spread to Rijeka and remained there for several hours. During that
time other rainfall bands formed and moved over Rijeka intensifying the precipitation and
prolonging the period of high precipitation intensity.
According to hourly amounts, precipitation intensity was the highest from 2100 to 2200 UTC (85.3
mm/h), with 20.6 and 51.7 mm/h in the previous and the next hour (Figure 9).

Sounding data measured at Zadar-Zemunik, located about 150 km southsoutheast of the area where
the largest rainfall was recorded, are shown to provide information on the vertical structure of the
troposphere. Although the thermodinamic profile characteristics are not completely representative
of the pre-convective environment over the study area, this is the only available sounding data on
the eastern Adriatic. The soundings featured a low-level moist atmospheric layer from the surface to
approximately 850 hPa connected with SE *jugo* wind, confirming a suitable environment for strong
convective activity (not presented). Winds strengthened throughout the troposphere and the highest
intensity was observed at 400 hPa.

**4.3. Operational model forecasts**


During the SOP1, DHMZ made available the operational forecast by ALADIN operational forecasts
model in 8 km and non-hydrostatic 2 km horizontal resolution (Section 2.3). A comparison between
two versions of ALADIN model is presented here and shows the capability in forecasting the
intense convective activity in the area.
Short-range forecasts reproduced well the large-scale and mesoscale features responsible for the
event (Figure 8). The low-level wind field is dominated by two low-level jet stream (LLJ) caused
the appearance of the low-level wind convergence over the North Adriatic and associated with the
main Genoa cyclone (Figure 8b). In this case the performance of the model is rather successful in
comparison with ECMWF reanalysis (not presented). One SW LLJ was elongated from Italy
towards the middle Adriatic that carry the warm and humid Mediterranean air to the Adriatic Sea,
and another NE LLJ (*bora* wind) modified and intensified by the pressure gradient across the
southern flank of the Alps (Figure 8a). This convergence was responsible for the convective
triggering in the late afternoon. Although the mesoscale characteristics are correctly reproduced,
the location and timing of precipitation was not so good. The intensive precipitation event was
predicted by both models with precipitation close or exceeding 100 mm/24 hours inland of Rijeka
(Figure 6), but the amount of precipitation was underestimated for the city of Rijeka that lies on the





coastline for all operational models possibly due to an absence of the cold pool that formed after the
showers in the early afternoon or low level wind from northeast that started earlier than in the
model forecast.
Operational forecast set-up of the ALADIN 2 km resolution run overestimates rainfall above
mountains (at least when compared to the 3B41 products from the TRMM data server), but it is
consequently closer to the extreme amounts measured in the Rijeka area (Figure 11). Although the
3B41 product is an estimate of precipitation intensity that also suffers from errors, the rain over the
southern Velebit Mountain was an overestimate, while it was correct for mountains inland of Rijeka.
In the hours of peak precipitation intensity in Rijeka, the satellite measurement data-derived
precipitation (TRMM 3B41RT product available from NASA's Giovanni web service) was also
considerably lower than the one measured in-situ.
The high resolution non-hydrostatic operational forecast shows upward motions along the coastal
mountains of Croatia and associated to the convergence line and the rain band over the sea (Figure
12). The wave of the upward motion moves from the Po valley eastward and reaches Rijeka area
one hour after the recorded maximum intensity in precipitations so the model might be little late
behind the real weather events. There is also a permanent wave formed over southern Velebit (and
several other mountains) that persist throughout the night. This wave is responsible for triggering
the precipitation there and its intensity is probably overestimated. Apparently, small but tall
topographic obstacles are able to trigger too much precipitation and this remains an issue to solve.

Figure 13 presents a scatter plot of 24h accumulated precipitation from rain gauges over Croatia and
forecast values from ALADIN model taken from the nearest grid points for IOP 2. ALADIN 8 km
model underestimated precipitation and forecasted up to 92 mm/24h of rainfall while measurements
reached 220 mm/24h. Much better results are obtained for ALADIN 2 km model where values
predicted by model were reached 200 mm/24h. A location error is also evident for both models
especially for the area where most intense precipitation occurred (Istria peninsula; red dots) but it is
smaller for ALADIN 2km model. Medium precipitation amounts are better forecast than strong one
but still slightly overestimated for ALADIN 8 km model and much more spread is noticeable for
ALADIN 2km model with both overestimation and underestimation but with better results for Istria
peninsula. From Table 2 and Table 3 it can be seen that ALADIN 2 km was relatively more accurate
(higher CSI) for dry and strong but not for medium category than ALADIN 8 km. FBIAS is better
for ALADIN 2 km for dry and strong but also for medium category compared to ALADIN 8 km
results.





**4.4 Influence of the data assimilation**

Since, the lack of model skill in simulating HPE may be partially attributed to imperfect initial
conditions, we perform several numerical weather prediction experiments to assess the impact that
data assimilation had on the IOP2 forecast accuracy.

Comparison of measurements with operational forecast and simulations without data assimilation is
shown in Figure 14. Rain gauges show that along Croatia-Slovenia border elongated area of
stronger precipitation is present and this pattern is better forecasted with operational run
incorporating data assimilation. Also over Istria peninsula higher amounts of medium rain category
are found in operational run which is in better accordance with measurements. This is also visible at
Figure 13 where for run with data assimilation points are less scattered and more points with higher
values of precipitation over Istria are present. Maximum recorded around the town Rijeka is not
adequately represented by either of the models.
Verification measures (Table 2) show that slightly better results are found for simulation with data
assimilation. Scores for the entire Croatia show that results in strong precipitation category are
improved for operational run (CSI=0.28) compared to run without data assimilation (CSI=0.23).
Also PCC shows that there is better association of model and observations for run with data
assimilation. Impact of data assimilation for this IOP is rather small but it still gives improvement in
24 hours precipitation forecast. It should be taken into account that for selected case better results
were obtained with higher resolution model and that data assimilated in operational ALADIN 8 km
model is mainly synoptic data. Thus, implementing data assimilation in higher resolution model and
adding additional high resolution temporal and/or spatial data to data assimilation system seems as
good way to further enhance operational forecast.


**Summary and conclusions**

In this paper an overview of the IOPs that affected the Adriatic TA during SOP1 HyMeX campaign
(5 September to 6 November 2012) is presented. During SOP1 20 IOPs were declared and 8 of
these events affected the EOP Adriatic TA. All of them produced localized heavy precipitation and
often were properly forecast by the available operational model ALADIN but exact prediction of the
amount, precise time and location of maximum intensity were missed. The total precipitations for
the SOP1 were above the corresponding climatology for the Adriatic TA. Maximum of precipitation
(more than 1.000 mm in 61 days at some locations) recorded on the northern Adriatic (city of



Rijeka) and its mountainous hinterland of Gorski Kotar. This region experiences climatic maxima of
the annual precipitation greater than 3.000 mm on average. Analysis was done mostly by the
measurements from the operational meteorological network maintained by the Meteorological and
Hydrological Service of Croatia.
There were 15 days when the accumulated rainfall on any of the raingauges in the Adriatic TA
exceeded 100 mm in 24 hours. Most the HPEs contain similar ingredients and synoptic setting but
of different intensity: a deep upper level through, cyclone strengthening over the Mediterranean (or
developing over Gulf of Genoa, Lyon or Tyrhennian sea), strong southwesterly low-level jet stream
that advects the moist and warm air towards the orographic obstacles along Mediterranean coastline
and destabilizes the atmosphere as the strong wind picks up the moisture from the sea.

Verification of the operational precipitation forecasts during SOP 1 suggests the operational
ALADIN at 8 km grid spacing model may be useful for early warnings to severe precipitation
events in the region. For most of the events there was high level of association between
precipitation forecast and measurements. From verification statistics and different precipitation
related figures it can be seen that one obvious limitation of ALADIN 8 km model is inability to
produce high amounts of precipitation and also tendency to underestimate frequency of dry events.
Having model at higher resolution (ALADIN 2 km) brings improvement for both problems but now
it slightly deteriorates forecast of medium precipitation and overestimates frequency of strong
precipitation events. Verification methods used in this work have their limitation where for
calculation of scores method used is point based comparison and thus it is prone to location error
and other methods that are used based on subjective comparison of different precipitation plots.
Next step would be implementation of object-based verification method e.g. SAL (Wernli et al.,
2008) which could provide more objective verification measures but for this local spatial
precipitation analysis must be developed first.

During the IOP2 on 12 September 2012, several thunderstorms formed including a supercell and a
possible tornado outbreak. The warm and moist air advected at the low levels over the Adriatic (and
Mediterranean before that) was feeding the storms, while apparently one storm produced
downdrafts that would in turn form a convergence zone with the moist flow from the sea and trigger
the next storm. Intensive precipitation event in Rijeka and surrounding area resulted from influence
of coastal mountains on the movement of a convergence line. The atmosphere contained a lot of
moisture, being close to saturated up to 6 km. The air flow converged above northern Adriatic in the
layer up to 2 km. The convergence line moved south-eastward. Rainfall intensified in Rijeka area
due to local terrain. The peak intensity was underestimated by the model forecast.




Such a chain of events poses a challenge with respect to predictability. The fact that the surrounding
mountains represent physical obstacles that modified the flow and determined the position of the
convergence zones made forecasting the location of such a chain of events easier. Abundance of real
time available measured data, including radar measurements, aircraft data and targeted radio
soundings can improve the initial conditions for the NWP models. The ambiguities in the sea
surface fluxes that pose an important source of energy for this event could be the factor that limits
the abilities of a deterministic forecast.

The numerical sensitivity experiments with respect to mesoscale data assimilation suggested the
precipitation forecast during IOP 2 was improved by using data assimilation to produce initial
conditions, compared to forecasts when initial conditions were derived from the global model data.
Use of mesoscale data assimilation for initial conditions enhances both precipitation structure and
intensity. This is evident also through improvement of objective verification measures, such as
critical success index and PCC. Data assimilation system could be further enhanced by using
additional observations (e.g. radar data, ground based GNSS data), shorter data assimilation cycle
(e.g. 3 hours instead 6 hour) or B matrix computed with different methods (ensemble B matrix
instead NMC based). Also work on implementing data assimilation system to higher resolution
model is ongoing.
Furthermore, operational non-hydrostatic model at 2 km grid spacing is able to predict the intensity
of a HPE more accurately than the hydrostatic model at 8 km grid spacing. Nevertheless, higher
resolution forecast can misplace the position of the peak precipitation and overestimate precipitation
over a narrow but high mountain such as southern Velebit. This may be an artefact of the excessive
sea surface temperature in the model in that region. These results suggest that precipitation forecast
in the Adriatic TA may be improved by both using mesoscale data assimilation and by decreasing
the grid spacing of the model.
Heavy precipitations over Adriatic area are often associated with sirocco (*jugo*) or *bora* winds, thus
involving intense air-sea interactions. In IOP4 was an excellent example for very intensive heat loss
caused by strong *bora* wind. In this case, control simulation run was more realistic with colder SST
and generally drier than the operational with a warmer SST. IOP4 shows the needs for further
improvements of the role of SST and surface (latent and sensible) heat fluxes over the Adriatic Sea,
which attain large values during strong Bora events. However a more detailed analysis of the impact
of SST on precipitation is ongoing.

Therefore, this paper highlights the need for enforcement an intensive observation period in the



future over the Adriatic region, to better understand the relevant processes and validate the
simulated mechanisms as well as to improve numerical forecasts via data assimilation and
improvements of model representation of moist processes and sea-land-atmosphere interaction.
There is also a need for collaborative effort within the Italian and other HyMeX scientific and
forecast communities to achieve a better understanding of the complex processes caused the
extreme events over the Adriatic region.

**Acknowledgements**
*This work is a contribution to the HyMeX program. The authors are grateful to the participating*
*institutions for providing the measured and model data. This work is partially supported by the Hymex-*
*COOP project (ENVIMED regional programme) and IPA2007/HR/16IPO/001-040510 grant. The authors*
*would also like to thank Jean-Francois Geleyn (deceased) the former project manager of ALADIN*
*for his ideas, energy, drive and persistence that made us an active party in developing a state of the*
*art model system and enabled us to participate in such important research programme. We thank*
*Marjana Gajić-Čapka for her precipitation extreme value analysis. We thank Iris Odak Plenković for*
*valuable advices and suggestion regarding precipitation verification. The authors are grateful to NASA for*
*providing valuable satellite derived products through the GIOVANNI web interface as well as TRMM, OMI*
*and MODIS scientists and developers.*

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

Hydrological Service: Zagreb





| Date | IOP | Location | Rainfall (mm) | Weather regime |
|---|---|---|---|---|
| 12-13 Sep | 2 | Rijeka | 220.2 | NAO+, cold front, SW advection |
| 13-14 Sep | 4 | Pelješac | 101.4 | NAO+, cyclone, bora and sirocco |
| 1–2 Oct | 9 | Rijeka | 127.4 | NAO+, cold front, SW advection |
| 11-13 Oct | 12a | Silba, Šolta, Prevlaka | 121.0 | blocking, cold front, SW advection |
| 14-16 Oct | 13 | Hvar, Mljet, Rijeka, Karlobag, Imotski | 118.6, 145.4 | blocking, cold front, SW advection |
| 26-28 Oct | 16 | Rijeka, Rijeka inland | 180.1, 173.5 | NAO-, blocking, cyclone, sirocco, aqua alta |
| 31Oct–2 Nov | 18 | Istria, Rijeka | 171.4 | NAO-, cyclone, sirocco, aqua alta |
| 4-5 Nov | 19 | Rijeka inland | 177.0 | NAO-, cyclone, SW advection |



























**Table 2:** *Verification measures calculated for 24 hour accumulated precipitation and for ALADIN 8 km*
*model (second column) for three categories (first column) and for whole SOP1 period (5 September to 6*
*November 2012), only IOP days (IOPavg) and for selected (IOP)s corresponding to time periods indicated*
*in Table 1 and for IOP2 without data assimilation experiment (IOP2 no DA). Verification measures include*
*Base Rate (BR), Frequency Bias (FBIAS), Critical Success Index (CSI) and polychoric correlation coefficient*
*(PCC). Due to zeros in contingency table some PCC scores could not be calculated (IOP4 and IOP16 for*
*ALADIN 8km model).*

| Cat. | Measure | Period | | | | | | | | | | |
|------|---------|--------|--------|------|---------------|------|------|--------|-------|-------|-------|-------|
| | | SOP1 | IOPavg | IOP2 | IOP2<br>no DA | IOP4 | IOP9 | IOP12a | IOP13 | IOP16 | IOP18 | IOP19 |
| Dry | BR [%] | 64.7 | 18.1 | 15.5 | 15.5 | 2.7 | 12.7 | 27 | 30.9 | 2.9 | 10.6 | 44.7 |
| | FBIAS | 0.78 | 0.29 | 0.5 | 0.41 | 0 | 0.15 | 0.47 | 0.45 | 0 | 0.01 | 0 |
| | CSI | 0.73 | 0.23 | 0.16 | 0.16 | 0 | 0.08 | 0.39 | 0.41 | 0 | 0.01 | 0 |
| Medium | BR [%] | 33.6 | 74.5 | 60.1 | 60.1 | 86.9 | 86.4 | 69.8 | 62.9 | 87.9 | 85.1 | 49.6 |
| | FBIAS | 1.45 | 1.2 | 1.36 | 1.39 | 1.03 | 1.1 | 1.24 | 1.26 | 1.09 | 1.14 | 1.91 |
| | CSI | 0.62 | 0.76 | 0.59 | 0.59 | 0.84 | 0.84 | 0.76 | 0.65 | 0.88 | 0.86 | 0.5 |
| Strong | BR [%] | 1.8 | 7.3 | 24.3 | 24.3 | 10.4 | 0.8 | 3.3 | 6.3 | 9.3 | 4.3 | 5.7 |
| | FBIAS | 0.63 | 0.73 | 0.42 | 0.42 | 0.98 | 3.75 | 0.19 | 1.13 | 0.42 | 0.69 | 0.89 |
| | CSI | 0.2 | 0.23 | 0.28 | 0.23 | 0.22 | 0 | 0 | 0.08 | 0.19 | 0.39 | 0.39 |
| | PCC | 0.8987 | 0.6847 | 0.5926 | 0.5488 | - | 0.3265 | 0.7489 | 0.7056 | - | 0.8824 | 0.7182 |

**Table 3:** *Same as Table 2 but verification measures are calculated for ALADIN 2 km model.*

| Cat. | Measure | Period | | | | | | | | | |
|------|---------|--------|--------|------|------|------|--------|-------|-------|-------|-------|
| | | SOP1 | IOPavg | IOP2 | IOP4 | IOP9 | IOP12a | IOP13 | IOP16 | IOP18 | IOP19 |
| Dry | BR [%] | 64.7 | 18.1 | 15.5 | 2.7 | 12.7 | 27.0 | 30.9 | 2.9 | 10.6 | 44.7 |
| | FBIAS | 0.92 | 0.81 | 0.83 | 1.69 | 1.29 | 0.76 | 0.74 | 0.79 | 0.64 | 0.84 |
| | CSI | 0.78 | 0.39 | 0.18 | 0.00 | 0.15 | 0.39 | 0.59 | 0.19 | 0.04 | 0.68 |
| Medium | BR [%] | 33.6 | 74.5 | 60.1 | 86.9 | 86.4 | 69.8 | 62.9 | 87.9 | 85.1 | 49.6 |
| | FBIAS | 1.12 | 1.00 | 1.11 | 0.85 | 0.86 | 1.12 | 1.07 | 0.98 | 1.01 | 1.09 |
| | CSI | 0.59 | 0.71 | 0.50 | 0.70 | 0.69 | 0.73 | 0.69 | 0.83 | 0.76 | 0.64 |
| Strong | BR [%] | 1.8 | 7.3 | 24.3 | 10.4 | 0.8 | 3.3 | 6.3 | 9.3 | 4.3 | 5.7 |
| | FBIAS | 1.65 | 1.49 | 0.84 | 2.08 | 10.75 | 0.38 | 1.64 | 1.22 | 1.76 | 1.46 |
| | CSI | 0.17 | 0.20 | 0.32 | 0.21 | 0.00 | 0.05 | 0.21 | 0.18 | 0.18 | 0.19 |
| | PCC | 0.8407 | 0.624 | 0.5302 | 0.3987 | 0.2083 | 0.4933 | 0.7896 | 0.3233 | 0.326 | 0.7854 |





**Table 4:** *Annual maximal precipitation amounts ($R_{max}$) recorded in different intervals t (minutes)*
*throughout the period 1958-2011 and during the heavy rainfall event on September 12, 2012 at*
*Rijeka and their return values (T) according to the GEV distribution applied to the period 1958-*
*2011.*

| t (minutes) | 1958-2011 | | 12 Sept 2012 | $T_{1958\text{-}2011}$ |
|---|---|---|---|---|
| | $R_{max}$ (mm) | T (year) | | |
| 5 min | 19.3 | 50 | 14.5 | 7 |
| 10 min | 29.2 | 54 | 24.6 | 12 |
| 20 min | 40.2 | 63 | **46.7** | >1000 |
| 30 min | 55.5 | 69 | **63.7** | 415 |
| 40 min | 67 | 48 | **74.8** | 130 |
| 50 min | 77.8 | 40 | **80.8** | 62 |
| 60 min | 86.4 | 40 | **87.4** | 43 |
| 120 min | 138.9 | 38 | **141.1** | 40 |
| 4 h | 194.9 | 80 | 171.8 | 52 |
| 6 h | 252.5 | 103 | 181.5 | 36 |
| 12 h | 317.3 | 214 | 200.9 | 37 |
| 18 h | 324.7 | 228 | 205.3 | 29 |
| 24 h | 324.7 | 232 | 208.3 | 25 |










**Figure 10:** *IR temperature enhanced satellite image for 2100 UTC on 12 Sep 2012 operational*
*MSG product used in DHMZ at the time.*
**Figure 11:** *High resolution forecast of hourly accumulated precipitation (shaded background) and*
*TRMM 3B41RT precipitation estimates (squares) for 1900 (a), 2000 (b), 2100 (c), 2200 (d) and*
*2300 (e) UTC 12 and 0000 (f) UTC 13 September 2012, this was the period of highest precipitation*
*intensity. Satellite derived precipitation data are used as provided from the Tropical Rainfall Measuring*
*Mission (TRMM, (Huffman et al. 2007)), in particular we used the hourly precipitation intensity data from*
*3B41RT product.*
**Figure 12:** *Vertical velocity omega (Pa/s) at 850 hPa level from the operational 2 km resolution*
*forecast for 2200 (a) and 2300 (b) UTC on 12 and 0000 (c) and 0100 (d) UTC on 13 September*
*2012, upward motions are shown in shades of red and downward in blue.*
**Figure 13:** *Scatter plot of 24h accumulated precipitation from rain gauges over Croatia and model*
*equivalent from ALADIN 8 km (left), ALADIN 8 km without data assimilation (middle), ALADIN 2*
*km (right) model and from the point nearest to the location of rain gauge for IOP2. With red colour*
*locations from Istria peninsula are marked.*

**Figure 14:** *24h accumulated precipitation from 12 Sep 0600 UTC until 13 Sep 0600 UTC (IOP12).*
*Left: rain gauge measurement, middle: ALADIN 8 km operational forecast with data assimilation,*
*right: ALADIN 8 km forecast without data assimilation.*






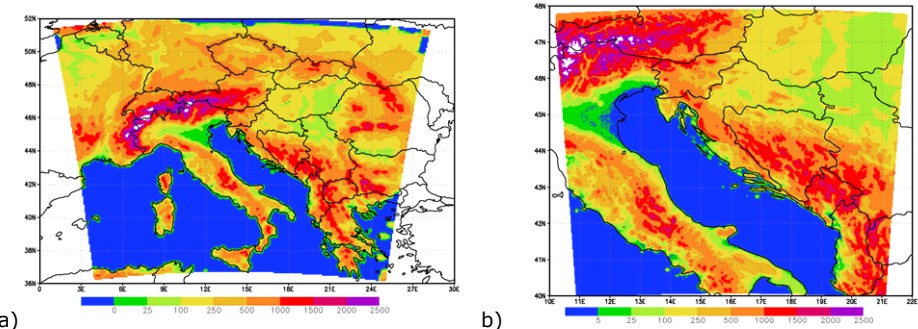

**Figure 1**: *ALADIN model domain and terrain height in 8 km (a) and 2 km (b) horizontal resolution.*




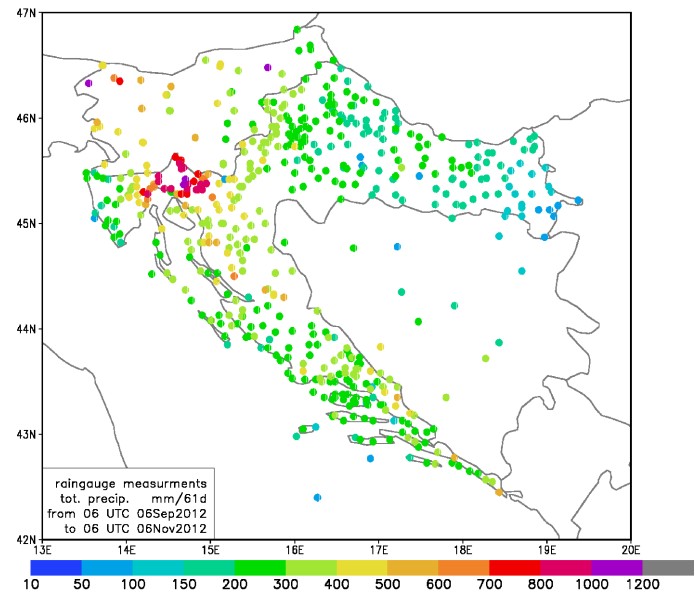

a)

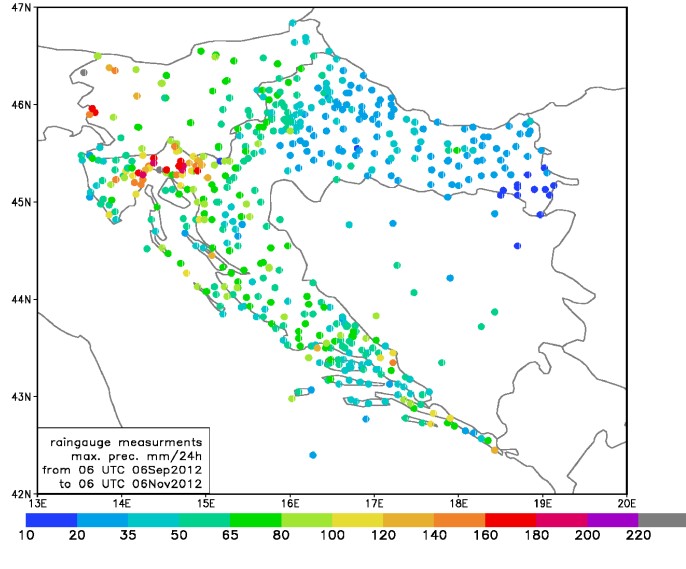

b)

**Figure 2:** *a) Total precipitation measured by the Croatian rain gauge network, cumulated over the whole SOP1 period; b) Maximum 24-h rainfall totals at each rain gauge station during the SOP1.*



**Figure 3**

Figure 3: Horizontal wind at 10 m (arrows coloured according to wind speed) and mean sea level pressure (blue isolines) forecasts by the ALADIN 8 km resolution run for 1200 UTC for: a) IOP4 (13 September); b) IOP9 (1 October); c) IOP13 (15 October); d) IOP16 (27 October); e) IOP18 (31 October); f) IOP19 (4 November). ☐




## Figure 4A

a)

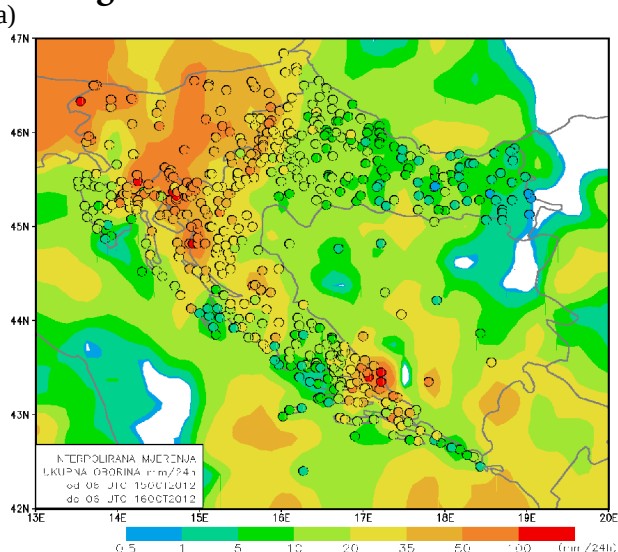

Figure 4: A - IOP13 (16 October): accumulated 24 hourly rainfall measured on rain gauges (circles) and interpolated
using data from rain gauges and a ccumulated 3B42RT 3 hourly product
for periods starting at 0600 UTC (a);
accumulated 24 hourly precipitation forecasts from the ALADIN 8 km resolution run (starting from 00 UTC on the same day)
(b) and for ALADIN 2 km resolution run (c).

b)

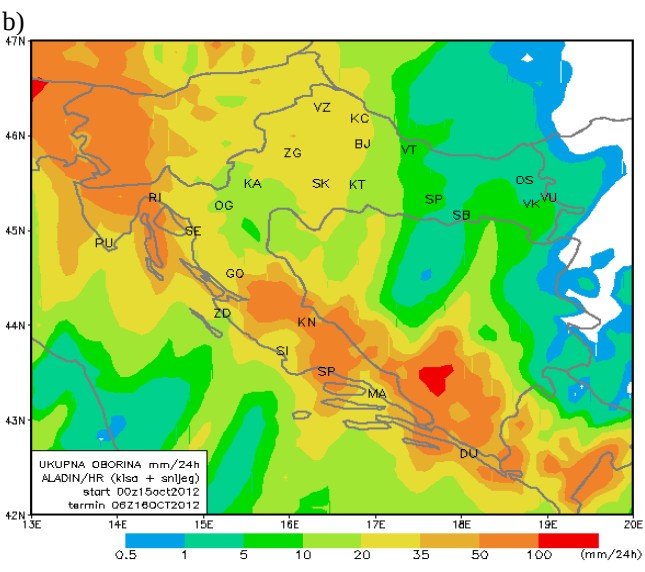

c)

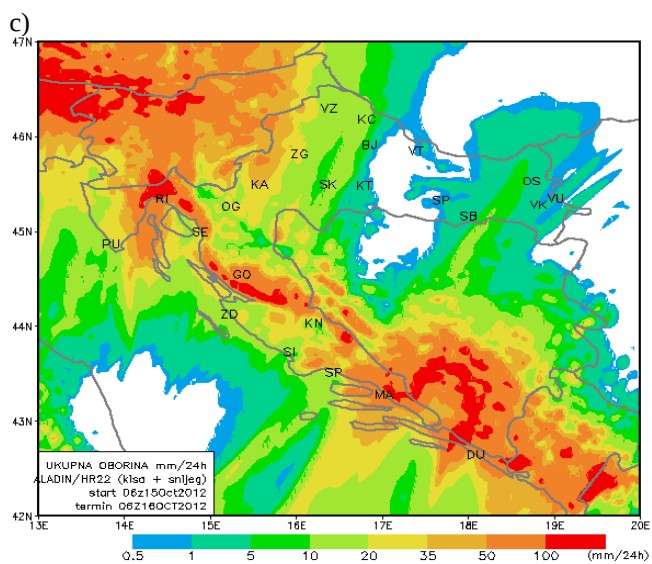



## Figure 4B

a)



*Figure 4*
*B: same as A but for IOP16 (28 October)* □

b)

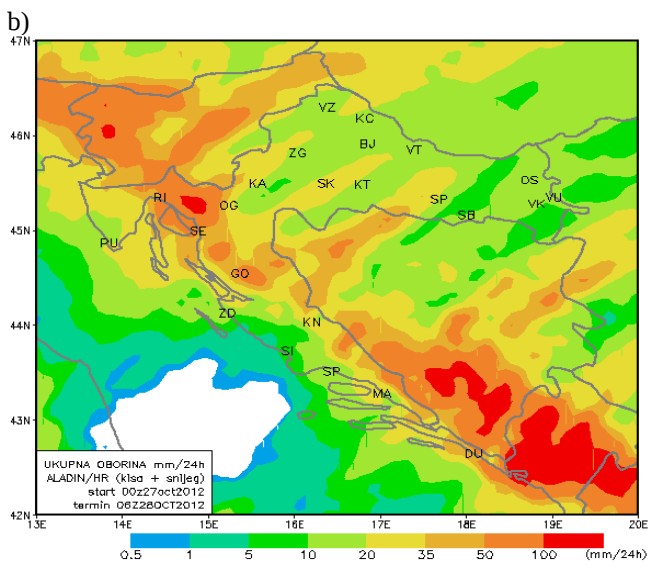

c)

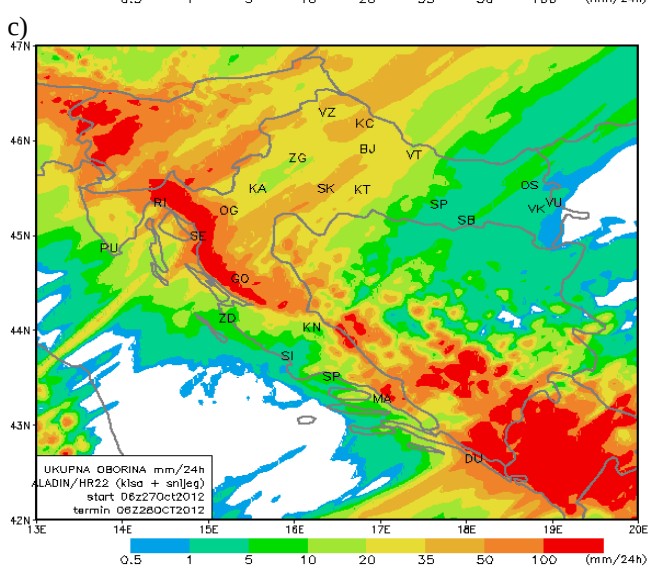




## Figure 4C

a)

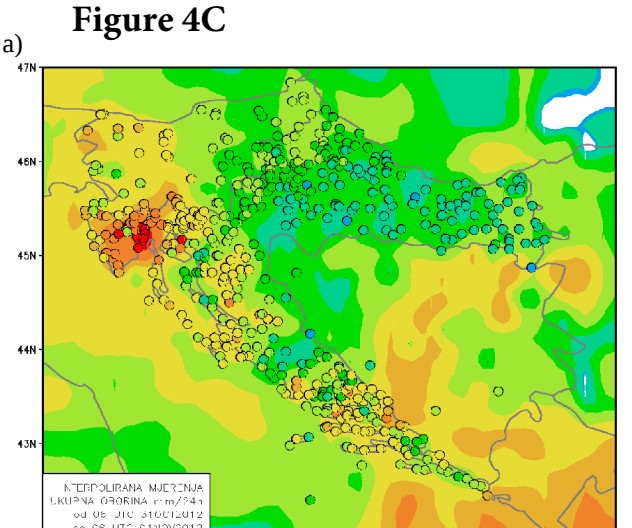

Figure 4
C: same as A but for IOP18 (1 November)

b)

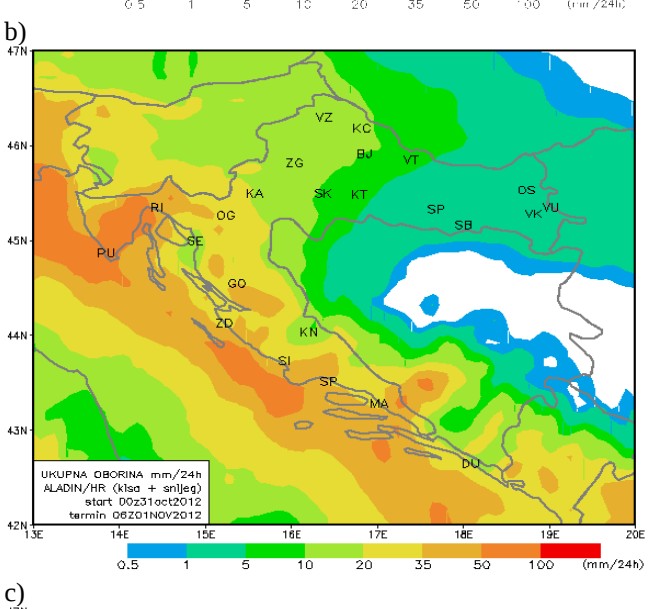

c)

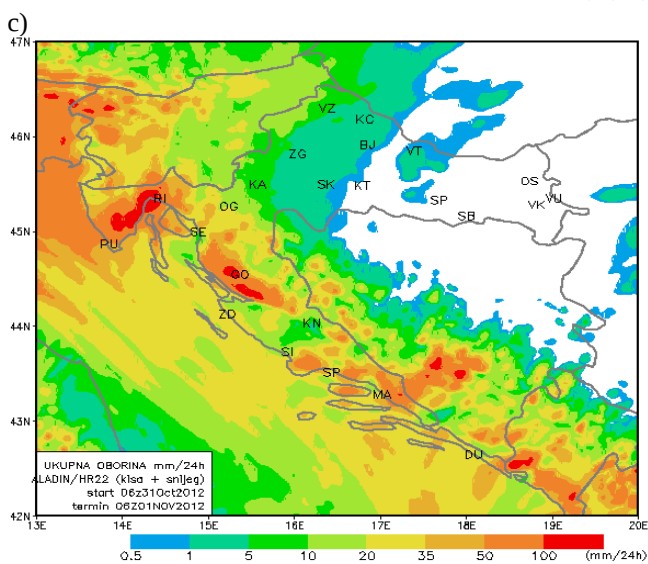


## Figure 4D

a)

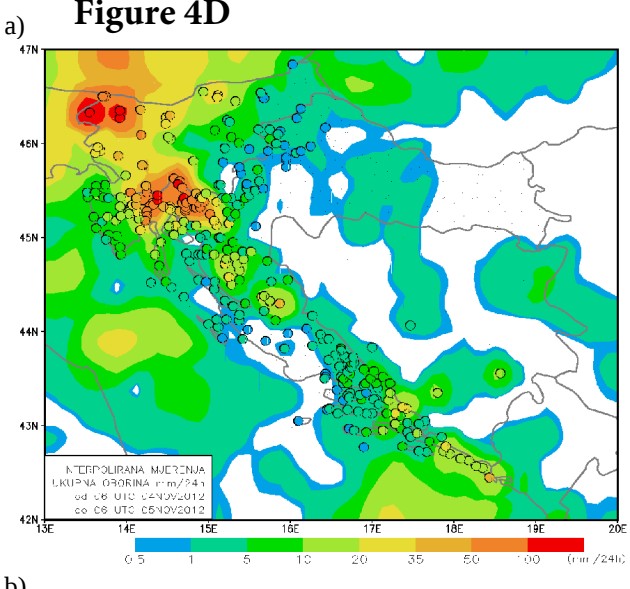

Figure 4
D: same as A but for IOP 19 (4 November)

b)

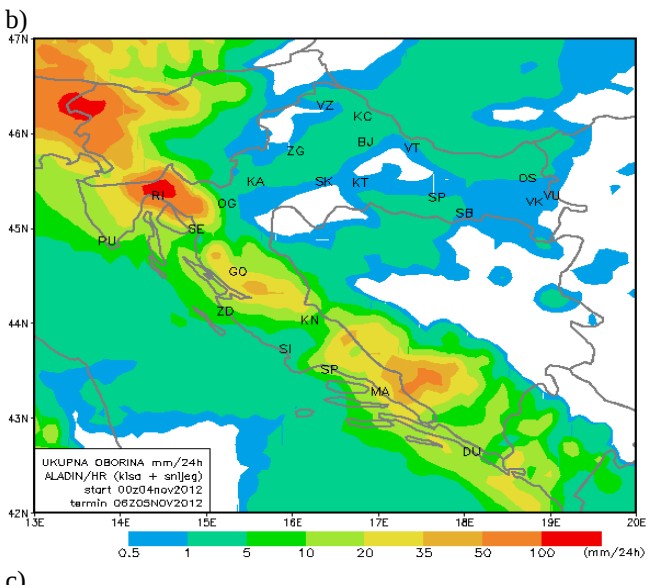

c)

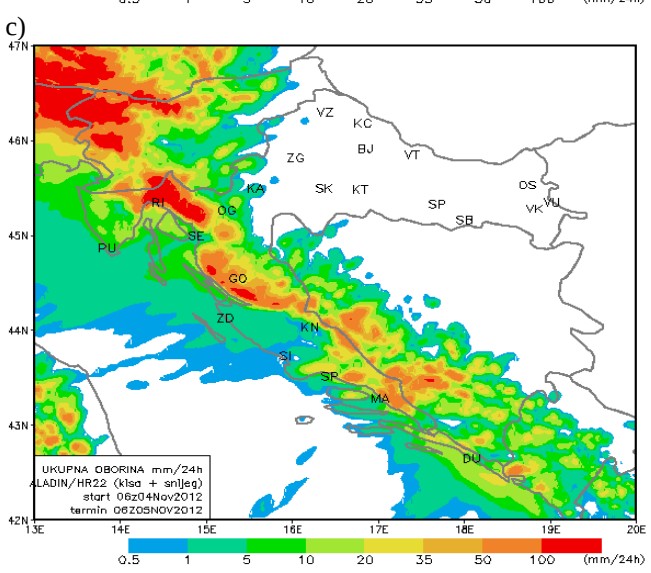





**Figure 5**

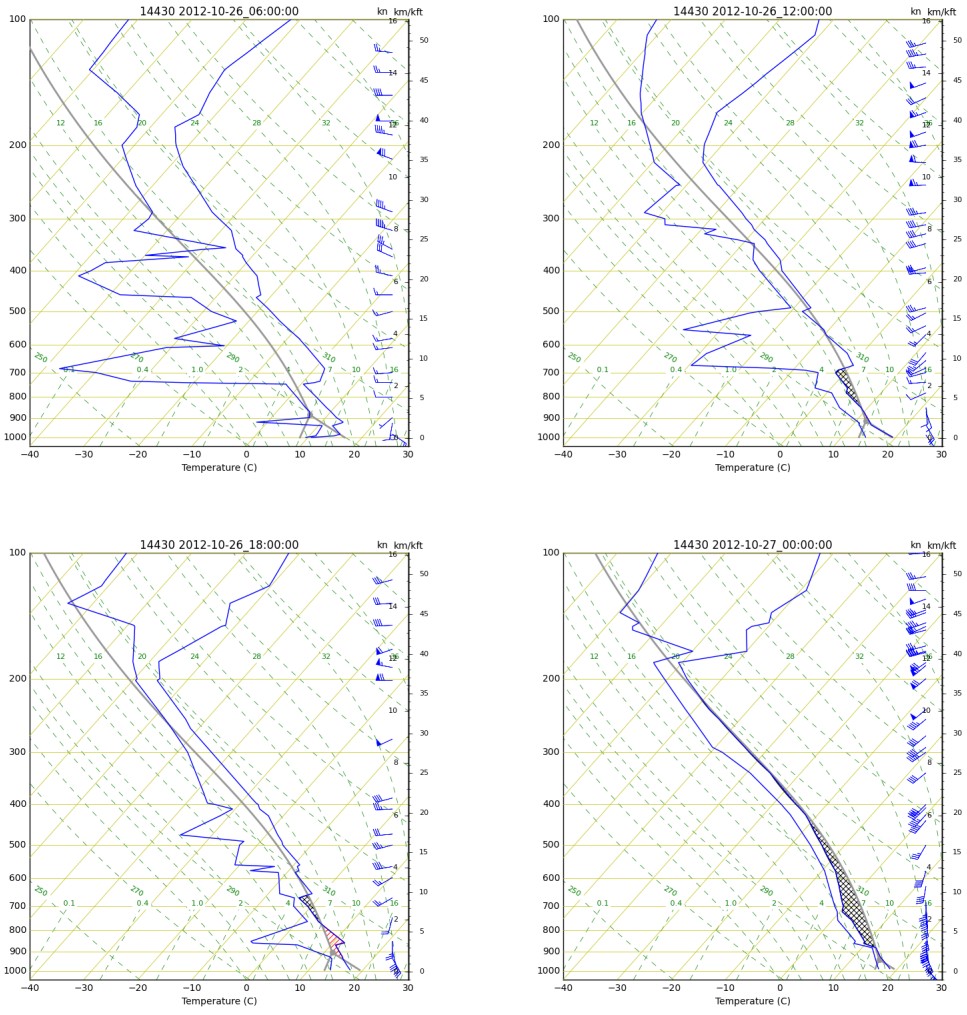

Figure 5: Radiosounding data for Zadar 26 October 2012 at 0600 and 1200 UTC (first row), 26 October 2012 at 1800 and 27 October 2012 at 0000 UTC (second row).





## Figure 6

*Figure 6: a) Sea surface temperature measured in situ (red) on station Bakar close to the city of Rijeka and the nearest sea point data used in the ALADIN 8 km resolution model from the global ARPEGE model (light blue) and OSTIA (blue) for the SOP1 from 5 September to 8 November 2012. For IOP4 (14 September) b) Accumulated 24 hourly rainfall measured on rain gauges (circles) and interpolated using data from rain gauges and 3B42RT 3 hourly product for periods starting at 0600 UTC; c) accumulated 24 hourly precipitation forecasts from the ALADIN 8 km resolution; d) accumulated 24 hourly precipitation forecasts from the ALADIN 2 km resolution run with SST from OSTIA; e) accumulated 24 hourly precipitation forecasts from the ALADIN 2 km resolution with SST from ARPEGE global model.*




Natural Hazards
and Earth System
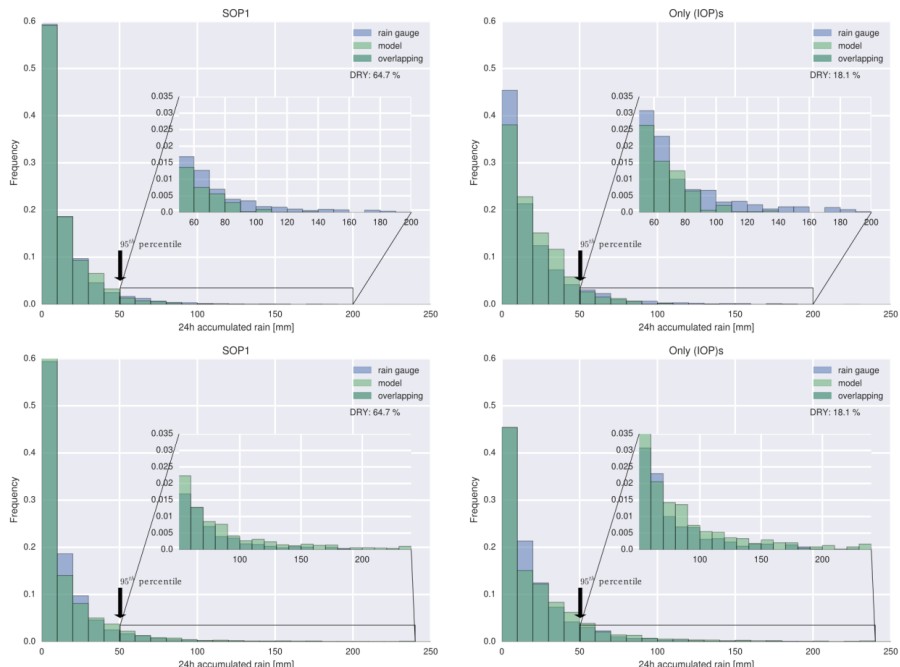

**Figure 7:** *Normalized histogram of rain events (accumulated precipitation on rain gauge station greater or equal 0.2 mm/24h) for the whole SOP1 period (5 September to 6 November 2012) (left column) and for days of selected (IOP)s within the same period (right column). In order to have readable histogram first histogram bin starts from 0.2 mm while number of dry days for given period is indicated at graph. Location of 95th percentile of SOP1 rain events distribution (50.42 mm/24h) is shown. Area of histogram after 95th percentile is zoomed and shown as inset to enhance readability. Frequency of precipitation events for rain gauge is coloured blue, for model light green, while dark green indicates overlapping of model and rain gauge data. First row: ALADIN 8km, Second row: ALADIN 2km upscaled to ALADIN 8km grid.*



# Figure 8

Figure 8: Mean sea level pressure (a) and 850 hPa geopotential height (blue isolines), wind speed (background shading) and direction (vectors) (b) according to the ALADIN model operational forecast on 2100 UTC 12 September 2012 (starting from the 0000 UTC analysis of the same day). □







**Figure 9**

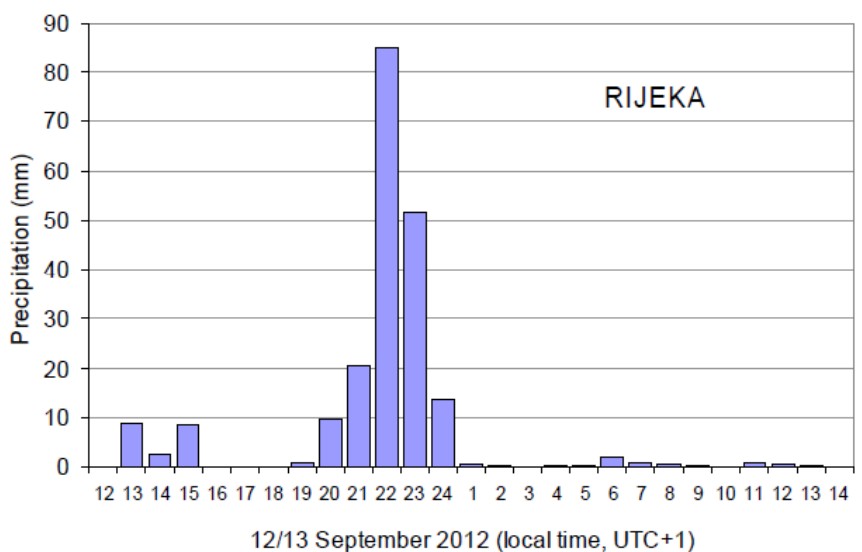

Figure 9: Hourly precipitation amounts from 1PM on 12 September 2012 to 1 PM on 13 September 2012 recorded at the Rijeka meteorological station.



**Figure 10**

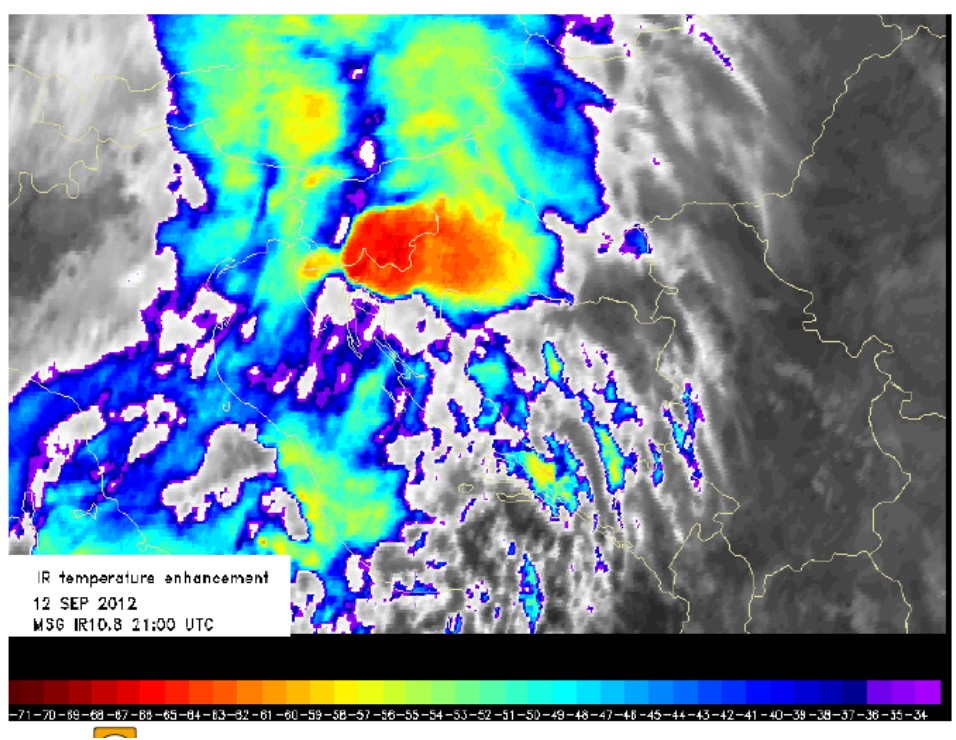

Figure 10: IR temperature enhanced satellite image for 2100 UTC on 12 September 2012 operational MSG product used in DHMZ at the time.





**Figure 11**



Figure 11: High resolution forecast of hourly accumulated precipitation (shaded background) and TRMM 3B41RT precipitation estimates (squares) for 1900 (a), 2000 (b), 2100 (c), 2200 (d) and 2300 (e) UTC 12 and 0000 (f) UTC 13 September 2012, this was the period of highest precipitation intensity. Satellite derived precipitation data are used as provided from the Tropical Rainfall Measuring Mission (TRMM, (Huffman et al. 2007)), in particular we used the hourly precipitation intensity data from 3B41RT product.



**Figure 12**

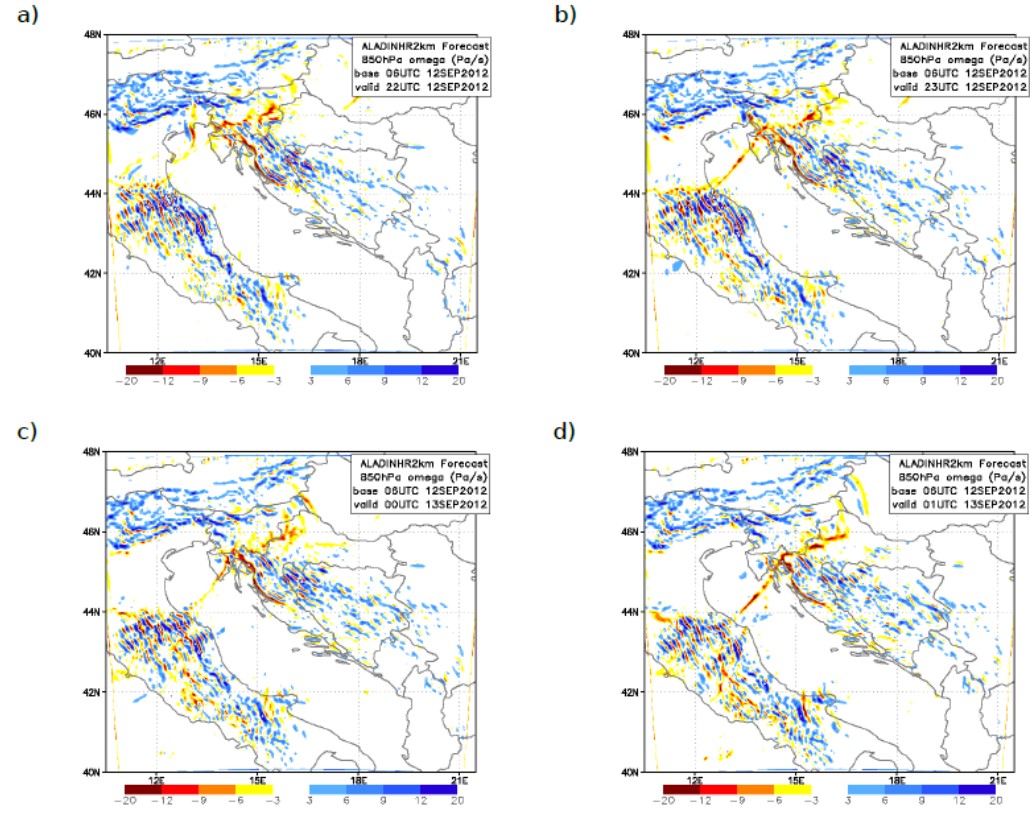

**Figure 12: Vertical velocity omega (Pa/s) at 850 hPa level from the operational 2 km resolution forecast for 2200 (a) and 2300 (b) UTC on 12 and 0000 (c) and 0100 (d) UTC on 13 September 2012, upward motions are shown in shades of red and downward in blue.**



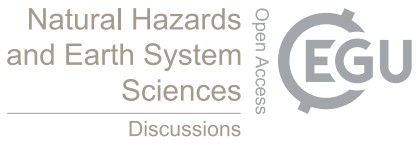
**Figure 13**

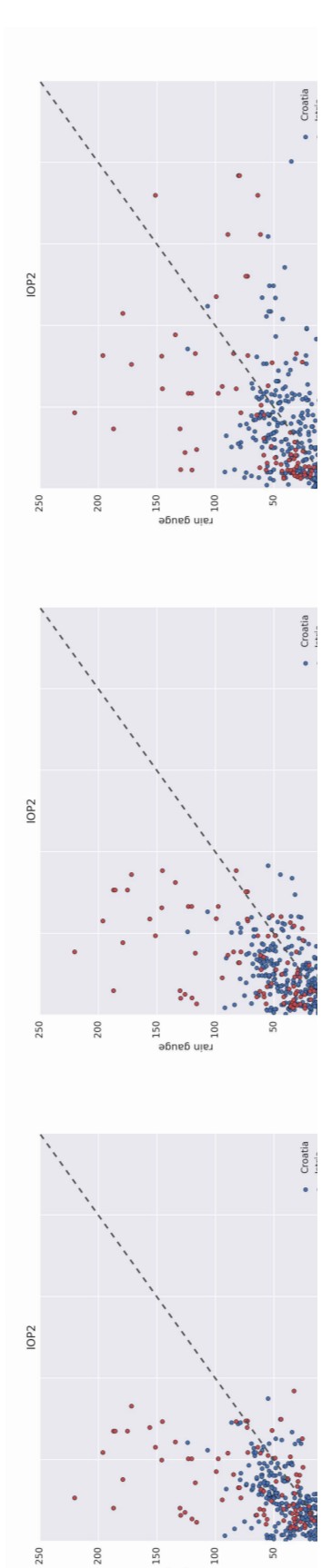

Figure 13: Scatter plot of 24h accumulated precipitation
from rain gauges over Croatia and model equivalent from
ALADIN 8 km (left), ALADIN 8 km without data assimilation (middle),
ALADIN 2 km (right) model and from the point nearest
to the location of rain gauge for IOP2.
Locations from Istria peninsula are marked with red colour.





*Figure 14: 24h accumulated precipitation from 12 Sep 0600 UTC until 13 Sep 0600 UTC (IOP12). Left: rain gauge measurement, middle: ALADIN 8 km operational forecast with data assimilation, right: ALADIN 8 km forecast without data assimilation.* □

**Figure 14**