# Peer review of "Overview of the first HyMeX Special Observation Period over Croatia"

_Natural Hazards and Earth System Sciences, 2016_

## Referee Comment (RC1) · Anonymous Referee #1 · 29 Aug 2016

Title: Overview of the first HyMeX Special Observation Period over Croatia
Authors: Ivančan-Picek et al.
RECOMMENDATION: Minor revisions

The high impact events occurred over Croatia during the first HyMeX SOP are discussed in this interesting paper, which nicely summarizes the outcomes and lessons learned from the participation of DHMZ in HyMeX. The paper is appropriate for NHESS, thus I recommend its publication after some relatively minor modifications.

Anyway, I strongly recommend that the authors should improve two points. First, I think the text would definitely benefit from a serious proof-read by a native English-speaker. Second, the organization of the paper should be improved. Actually, the reader may have troubles following the description of the different IOPs, since the authors jump from one figure to another and it is difficult to follow the information flow (this is true in particular for Subsection 3.1). The improvement I ask for is necessary for a better readability of the paper in order to convince the interested reader that is worth to read until the end. With this purpose, it may be convenient also to reduce the length of the manuscript and, possibly to remove some figure panels.

Minor points:
Line 139-141: I think it would be beneficial to add the locations of radiosounding stations and radar sites in Fig. 2;
Line 144: … majority…: how many?
Line 152: The dense network of climatological stations …: how many? What is the horizontal resolution of the network?
Line 153: why are the synoptic observations not taken at the main synoptic hours?
Lines 165-167: It is not clear what SAP refers to: is it a technique to select relevant parameters?
Line 199: why is the convection parameterization employed at 2 km grid spacing? Why not using an explicit treatment?
Subsection 2.3.1 is too long and provides unnecessary details; it should be strongly reduced;
Line 218: what is the biperiodization?
Line 312-316: the details about NAO are unnecessary since it is a well-known index;
Line 364: where is Rijeka? A map with the location of the places mentioned in the text should be added (maybe using Fig. 2);
Lines 390-391: sentence not clear;

Line 434: add a sentence like "Especially in a narrow and inhomogeneous basin like the Adriatic, small-scale SST variations cannot be properly represented in the coarse large-scale analysis, especially near the coasts";

Line 459: "…less **than** or equal **to** 0.2 …"; is this calculated in each grid point?

Line 471-499: the statistical analysis requires the definition of the indices used in Tables 2 and 3; this could be done in an Appendix;

Section 4: since the Section is completely dedicated to IOP2, it may be convenient to add the reference to Miglietta, M. M., Manzato, A. and Rotunno, R. 2016. Characteristics and Predictability of a Supercell during HyMeX SOP1. Q.J.R. Meteorol. Soc., doi:10.1002/qj.2872, which focuses on the convective episodes of IOP2 over northeastern Italy;

Figure 6: it seems like ARPEGE has a systematic bias in SST: what is its resolution?

Other points:

Line 53: rephrase in this way "… data assimilation especially at the convective scale. HyMeX …";

Line 55: …responsible **for** their …

Line 95: … on the southe**astern** part of …

Line 99: … expanded **to** the Adriatic …

Line 125: including -> included

Line 192-193: … the operational **forecasting system** (Tudor et al., 2013). At **that** time …**,** run twice per day on a domain **with 8 km grid spacing** (Fig. 1a) …

Line 266: … **an** additional operational forecast run …

Line 286: 0000 UTC instead of 0600 UTC;

Line 300: rephrase "Similar results were found over the Apennines in the Italian peninsula …"

Line 309: "as a favorable condition to …"

Line 367: "Occasionally, a mesoscale cyclone **developed,** associated with …";

Line 367: do you refer to low or upper level PV anomaly?

Line 369: "**The** mesoscale cyclone moved …"

Line 373: "over southeast coast of the Adriatic **in Croatia** …"

Line 378: "This weather regime **(Fig. 3b)** …"

Line 383: "Smooth troughs **(Fig. 3c)** entering …"

Line 384: "a **southwesterly low-level**(?) flow over the Adriatic TA"

Line 397: "A **southwesterly** flow **(Fig. 3f)** …"

Line 398: "… that formed **between** northwest Europe **and** northern Africa …"

Line 402: "… orographic**ally** forced …"

Line 410: "**A** mesoscale cyclone or a frontal system **moving** slowly southeastward …"

Line 413: "… southeast Adriatic coast **of Croatia** and …"

Line 436: "… the SST decreased **by** 10°C **in the** station **of** Bakar …"
Line 442: "… the simulation **with modified SST** is more realistic **(cf. with Figure 6b)** …"
Line 445: "… improvements **due to** the role …"
Line 498: IOP9 or IOP19?
Line 518: "…southwesterly flow…";
Lines 524-526: "During **late afternoon** … resulted **in** intensive convective processes…"
Line 534: "a more" instead of "the more";
Lines 541-543: "… in more than thousand … belong to the category of extraordinary rare events ... expected once in forty …"
Line 573: "data … are shown …": really, they are not shown;
Line 589: "… jet stream**s** …";
Line 592: "(not presented)"; really, they are actually presented, but the caption does not describe these panels;
Line 594: "… NE LLJ (bora wind)**,** modified and intensified by the pressure gradient**,** across …";
Line 613: "The **line of** upward motion moves …";
Line 615: "…permanent **uplift** over …";
Line 625: "…**the** most intense precipitation …";
Line 655: "…selected case**s**…";
Line 658: remove additional (repetition); "…seems **a** good way…";
Line 669: "Maximum precipitation **was** … recorded …"
Line 688: remove "now";
Line 690-692: "… since the scores are based on point comparison, thus it is prone to location and other errors based on …"
Line 694: "… but**,** for this**,** local spatial precipitation …"
Line 712: "…**a** factor …"
Line 729: "… in that region. **Results** suggest …"
Line 733: "interactions. IOP4 …"
Line 745: "… processes **that** caused …"
Figure 3 caption: "…**at** 1200 UTC…"
Figure 7 caption: "… greater than or equal to … frequency of dry days for a given period …"
Figure 9 caption: "Hour**ly** precipitation …"
Figure 13 caption: "Red colors are used for locations in Istria peninsula".

---

## Author Comment (AC1) · 1 Sep 2016

Reply to reviewer comments on "Overview of the firs HyMeX Special Observation Period over Croatia" by Ivančan-Picek, Tudor, Horvath, Stanešić and Ivatek-Šahdan

The authors would like to thank the Reviewer for the through review of the manuscript. We will do our best to improve the manuscript, according to the comments. General comments: 1. We agree with the reviewer that that the manuscript would be more readable if English language native speaker would proof-red the manuscript and correct the grammar. Before submitting a revised final version of our paper, English text will be corrected by a native English speaker. 2. Accepted. We appreciate the comments made by the Reviewer, which pointed that the description of the different IOPs is difficult to follow in the information flow (particularly in Subsection 3.1). In the revised

manuscript we will remedy the problem. What we plan to do is to reorganise the text in accordance to the reviewer comments. In the Subsection 3.1 we will also highlight the different physical proceses that produced HPE during the different IOPs. Minor points: 1. Line 139-141; Line 364 – Locations of radiosounding stations, radar sites and other places mentioned in the text will be added in Fig. 2 2. Line 144: Majority of SYNOP stations are also equipped with an automatic station ... how many? We propose to change the sentences in Line 143-145 in: The meteorological measurements and observations on 58 SYNOP stations (31 of them are automatic stations) are done every hour and reported in real time during the SOP1. 3. Line 152: The number of climatological stations of the network in Croatia is 120. Average distance between stations are 20 km. We will add this information in the text. 4. Line 153: why are the synoptic observations not taken at the main synoptic hours? Our high- resolution analysis are based on the dense network of climatological stations that make the observations three times a day (06, 13 and 20 UTC). 5. Lines 165-167: It is not clear what SAP refers to: is it a technique to select relevant parameters? Sensitive area prediction is a prediction of where might a more accurate definition of the initial state of the atmosphere benefit the quality of the forecast over the region in question. Sensitive areas are regions where extra observations are expected to have the largest impact on the forecasts for the verification area.

We reformulated the sentence accordingly: The selection of sensitive area predictions (SAP), that is predictions of regions where observations are expected to have the largest impact on the forecasts for the verification, used methods developed by ECMWF and Meteo-France (Prates et al., 2009). 6. Line 199: Why is the convection parameterization employed at 2 km grid spacing? Why not using an explicit treatment? As explained in the text and more elaborately in references that describe the 2km resolution operational forecast and its parametrisations in more detail: ALADIN is a spectral model and operationally we are using quadratic truncation. This means that gridpoint resolution is 2 km but the shortest resolved wave has a wavelength of 6 km. The 3MT convection scheme can be run in multiple scales and substantial amount of literature

shows that substantial part of convection remains unresolved even in 1km resolution (e.g. Kajikawa et al., 2016). Therefore, we will add the reference: "Kajikawa et al., 2016: resolution dependence of deep convections in a global simulation from over 10-km to sub-kilometer grid spacing. Progress in Earth and Plnetary Science, DOI: 10.1186/s40645-016-0094-5"

7. Accepted. Subsection 2.3.1 is devoted to the description of the well known operational 8 km ALADIN forecast. Therefore, we will reduce the lenght of this section and remove unnecessary details which could find in the listed references. 8. Line 218: What is biperiodization? The biperiodization is a numerical technique to facilitate spectral computations for dynamics in LAM Specific for spectral LAM uses FFT. 9. Line 312-316: We agree. The details about NAO will be removed. 10. Line 390-391: Instead of sentence "Large-scale conditions such as found in these IOPs help to generate mesoscale and local processes which modify additionally flow regimes leading to quite different precipitation patterns" we propose "Similar large-scale conditions such as found in these IOPs help to generate mesoscale and local processes leading to quite different precipitation patterns" 11. Line 434: Accepted. We will add proposed sentence. 12. Line 459: No. To clarify this we propose to include in the text: ALADIN model at 2km grid spacing during SOP1 was assessed by comparing forecasts from the nearest model point with respect to the observation location with the measurements from Croatian surface observation network. 13. Line 471-499: We agree. The definition of the verification measures (indices) used in Tables 2 and 3 will be done in Appendix. 14. We appreciate the comments made by the Reviewer, which reminded the authors to the reference Migletta et al. (2016). We will refer to this paper which focuses on the IOP2 over northeastern Italy. 15. Figure 6 - What is ARPEGE resolution? In 2012, ARPEGE resolution over the western Mediterrannean Sea was about 11 km and more than 14 km easward (stretched grid). This is gridpoint resolution since ARPEGE is also a spectral model.

Other points:

All accepted and problem will be corrected.

Please also note the supplement to this comment:
http://www.nat-hazards-earth-syst-sci-discuss.net/nhess-2016-247/nhess-2016-247-AC1-supplement.pdf

---

## Referee Comment (RC2) · Anonymous Referee #2 · 6 Sep 2016

Title: Overview of the first HyMeX Special Observation Period over Croatia
Authors: Ivančan-Picek et al.
RECOMMENDATION: Major revisions

The authors present a review of the high impact events occurring in Croatia during SOP1 of the HyMeX campaign. In this paper an analysis of the main meteorological features of the events and the ability of the operational forecast are presented as well as the most important findings of the campaign.
Unfortunately, the paper lacks in clearly presenting the events making the readability quite low and referring to previous publications concerning the campaign in the introduction. Moreover, a proof-read by native English speaker would improve the readability of the paper.
Finally, plagiarism is not allowed, the two following sentences are the same as in Ferretti et al., 2014:
*The characteristics of the Mediterranean region, a  nearly closed basin surrounded by highly urbanized and complex terrain close to the coast, makes Mediterranean area prone to natural hazards related to the water cycle,….*

*Several events were characterised by convection over the sea followed by orographic precipitation……....*

My suggestion is to accept the paper for publication after a major revision.

General Comments:
Section 2 should be shortened (details on the observations and models can be summarized in a table).
The presentation of the events (section 3) is difficult to read. There is not correspondence between the figures presentation and their citation in the text. Moreover, the rationale with which the events are presented is not clear. This section should be rewritten and please show only the figure that are discussed in the text.

Specific Comments:
Pag. 2, line40-42: either rewrite the sentence or refer to Ferretti et al., 2014
Pag. 3 Line 98: ´Although the Adriatic TA…´Where is it? Please add a figure showing the TAs
Pag. 4, line 138: ´ DHMZ deployed a ground observation operational network…´ Please add a figure with the location of the observations (radar, soundings etc.,)  and special equipment if any.
Pag. 7, line 226-231 there is no need for a detailed description, refer to previous publications.
Pag. 10, line 316: previous paper have already assessed the role of Hurricane Nadine and NAO please add references (Pantillon et al., 2015, Ferretti et al., 2014).
Pag.10, line 351: figure 5 is called before then figure 4 (see general comments on this section)
Pag. 12, line 393: rewrite this sentence.
Pag. 14, line 464: ´….. during the whole SOP1 period and during the specific days corresponding to 8 IOPs …´ Specify the IOPs.
Pag.18, line 605: Figure 11 is not clear, are the squares showing the precipitation in color? If yes it is difficult to distinguish from the shaded background.
Pag.19, line 640: data assimilation of which data? Please add a sentence.

---

## Author Comment (AC2) · 29 Sep 2016

Reply to reviewer 2 comments on "Overview of the first HyMeX Special Observation Period over Croatia" by Ivančan-Picek, Tudor, Horvath, Stanešić and Ivatek-Šahdan

We appreciate the thorough review by the Reviewer and have done our best to improve the manuscript, according to the comments. We agree with the reviewer that that the manuscript would be more readable if English language native speaker would proof-red the manuscript and correct the grammar. Before submitting a revised final version of our paper, English text will be corrected by a native English speaker. We appreciate the comments made by the Reviewer, "the paper lacks in clearly presenting the events making the readability quite low". In the revised manuscript we will remedy the problem. What we plan to do is to reorganise the text in accordance to the reviewer

comments. Regarding the Reviewer comment that the two sentences are the same as in Ferretti et al., 2014, we are very sorry for that and confirm that this is accidental. During our work on this manuscript we consulted a lot of relevant references (many are cited in the paper) in which we found similar sentences construction. The content of these two sentences is general description of the Mediterranean region and well known convection as major source of heavy precipitation over the sea, and therefore does not have any influence on the presented results. In the revised manuscript we will rewrite the mentioned sentences in our own words.

General comments: Accepted. We appreciate the comments made by the Reviewer, which pointed that the description of the observations and models should be shortened. In the revised manuscript we will remedy the problem. What we plan to do is to remove unnecessary details and summarize details on observations and models in a separate table. We agree with the Reviewer that the presentation of the events in the Section 3 is difficult to read. Before submitting a revised final version of our paper, this section will be rewritten in accordance to the reviewer comments.

Specific Comments: Line 40-42: Accepted. We will rewrite the sentence. Line 98: To explain where is Adriatic TA we will refer to the HyMEX (www.hymex.org/?page=target_areas) where identified 3 main Mediterranean target areas: North-West (NW), Adriatic (A) and South-East (SE). Line 138: Agreed. We will add a figure with the location of the observations in Croatia. Line 226-231: Accepted. As we already noticed, Section 2 will be shortened. Line 316: Accepted. We will add suggested references and remove details about the NAO. Lines 351: Acknowledged. In the revised manuscript we will remedy the problem Line 393: Agreed. Modified. Line 464: Accepted. We will specify the IOPs. Line 605: Accepted. The squares show the precipitation. We will prepare new Figure 11 where the squares should distinguish from the shaded background. Line 640: We agree. The information about the data used in the data assimilation will be added.

Please also note the supplement to this comment:
http://www.nat-hazards-earth-syst-sci-discuss.net/nhess-2016-247/nhess-2016-247-AC2-supplement.pdf

---

## Author Response (AR1)

Reply to reviewers comments on

**„Overview of the first HyMeX Special Observation Period over Croatia"**

by Ivančan-Picek, Tudor, Horvath, Stanešić and Ivatek-Šahdan

We appreciate the thorough and detailed review, with useful suggestions. We have done our best to improve the manuscript in a considerable number of corrections and modifications, according to the Reviewers comments. We have been asked to make major revisions mainly in the language and presentation.

- English has been corrected and red by a native English speaker. Language proofreading certificate is attached.
- In the revised manuscript, we agree with both Reviewers and have reformulated Section 3. Additionally, we heavily shortened Section 2 and removed unnecessary details. In order to improve readability of the manuscript, we arrange the figures. Instead of previously Figure 4 A, B, C and D, in the revised version we have Figure 5, 6, 7 and 9.

**Reply to Reviewer #1 comments:**

The authors would like to thank the Reviewer for the through review of the manuscript. We have done our best to improve the manuscript, according to the comments.

**General comments:**

1. We agree with the reviewer that that the manuscript would be more readable if English language native speaker would proof-red the manuscript and correct the grammar. English text has been corrected by a native English speaker.
2. Accepted. We appreciate the comments made by the Reviewer, which pointed that the description of the different IOPs is difficult to follow in the information flow (particularly in Subsection 3.1). In the revised manuscript we remedy the problem. We reorganise the text in accordance to the reviewer comments. In the Subsection 3.1 we also highlight the different physical proceses that produced HPE during the different IOPs.

**Minor points:**

1. Line 139-141; Line 364 – Locations of radiosounding stations, radar sites and other places mentioned in the text added in Fig. 1b
2. Line 144: Majority of SYNOP stations are also equipped with an automatic station ... how many? We change the sentences in Line 143-145 in: *The meteorological measurements and observations on 58 SYNOP stations (31 of them are automatic stations) are done every hour and reported in real time during the SOP1.*

3. Line 152: The number of climatological stations of the network in Croatia is 120. Average distance between stations are 20 km. We add this information in the text.

4. Line 153: why are the synoptic observations not taken at the main synoptic hours? Our high- resolution analysis are based on the dense network of climatological stations that make the observations three times a day (06, 13 and 20 UTC).

5. Lines 165-167: It is not clear what SAP refers to: is it a technique to select relevant parameters?
Sensitive area prediction is a prediction of where might a more accurate definition of the initial state of the atmosphere benefit the quality of the forecast over the region in question. Sensitive areas are regions where extra observations are expected to have the largest impact on the forecasts for the verification area.

We reformulated the sentence accordingly:
*The selection of sensitive area predictions (SAP), that is predictions of regions where observations are expected to have the largest impact on the forecasts for the verification, used methods developed by ECMWF and Meteo-France (Prates et al., 2009).*

6. Line 199: Why is the convection parameterization employed at 2 km grid spacing? Why not using an explicit treatment?

As explained in the text and more elaborately in references that describe the 2km resolution operational forecast and its parametrisations in more detail: ALADIN is a spectral model and operationally we are using quadratic truncation. This means that gridpoint resolution is 2 km but the shortest resolved wave has a wavelength of 6 km. The 3MT convection scheme can be run in multiple scales and substantial amount of literature shows that substantial part of convection remains unresolved even in 1km resolution (e.g. Kajikawa et al., 2016).
Therefore, we add the reference: "*Kajikawa et al., 2016: resolution dependence of deep convections in a global simulation from over 10-km to sub-kilometer grid spacing. Progress in Earth and Planetary Science, DOI: 10.1186/s40645-016-0094-5*"

7. Accepted. Subsection 2.3.1 is devoted to the description of the well known operational 8 km ALADIN forecast. Therefore, we reduce the lenght of this section and remove unnecessary details which could find in the listed references. Details of the operational model characteristics are summarized in Table 1.

8. Line 218: What is biperiodization? The biperiodization is a numerical technique to facilitate spectral computations for dynamics in LAM Specific for spectral LAM uses FFT.

9. Line 312-316: We agree. The details about NAO are removed.

10. Line 390-391: Instead of sentence *„Large-scale conditions such as found in these IOPs help to generate mesoscale and local processes which modify additionally flow regimes leading to quite different precipitation patterns"* we propose *„Similar large-*

*scale conditions such as found in these IOPs help to generate mesoscale and local processes leading to quite different precipitation patterns"*

11. Line 434: Accepted. We add proposed sentence.
12. Line 459: No. To clarify this we propose to include in the text: *ALADIN model at 2km grid spacing during SOP1 was assessed by comparing forecasts from the nearest model point with respect to the observation location with the measurements from Croatian surface observation network.*
13. Line 471-499: We agree. The definition of the verification measures (indices) used in Tables 2 and 3 have done in Appendix.
14. We appreciate the comments made by the Reviewer, which reminded the authors to the reference Migletta et al. (2016). We refer to this paper which focuses on the IOP2 over northeastern Italy.
15. Figure 6 - What is ARPEGE resolution? Figure 6 in the revised version of manuscript become Figure 4. In 2012, ARPEGE resolution over the western Mediterrannean Sea was about 11 km and more than 14 km easward (stretched grid). This is gridpoint resolution since ARPEGE is also a spectral model.

**Other points:**

All accepted and problem corrected.

**Reply to Reviewer #2 comments:**

We appreciate the thorough review by the Reviewer and have done our best to improve the manuscript, according to the comments.

We agree with the reviewer that that the manuscript would be more readable if English language native speaker would proof-red the manuscript and correct the grammar. English has been corrected and red by a native English speaker.

We appreciate the comments made by the Reviewer, *„the paper lacks in clearly presenting the events making the readability quite low".* In the revised manuscript we remedy the problem and reorganise the text in accordance to the reviewer comments.

Regarding the Reviewer comment that the two sentences are the same as in Ferretti et al., 2014, we are very sorry for that and confirm that this is accidental. During our work on this manuscript we consulted a lot of relevant references (many are cited in the paper) in which we found similar sentences construction. The content of these two sentences is general description of the Mediterranean region and well known convection as major source of heavy precipitation over the sea, and therefore does not have any influence on the presented results. In the revised manuscript we rewrite the mentioned sentences in our own words.

**General comments:**

Accepted. We appreciate the comments made by the Reviewer, which pointed that the description of the observations and models should be shortened. In the revised manuscript we remedy the problem. We remove unnecessary details on observations and summarize models details in a separate table.

We agree with the Reviewer that the presentation of the events in the Section 3 is difficult to read. This section was rewritten in accordance to the reviewer comments.

**Specific Comments:**

Line 40-42: Accepted. We rewrite the sentence.

Line 98: To explain where is Adriatic TA we refer to the HyMEX (www.hymex.org/?page=target_areas) *where* identified 3 main Mediterranean target areas: North-West (NW), Adriatic (A) and South-East (SE).

Line 138:  Agreed. We add a figure with the location of the observations in Croatia (Figure 1b).

Line 226-231: Accepted. We reduce the lenght of this section and remove unnecessary details which could find in the listed references. Details of the operational model characteristics are summarized in Table 1.

Section 2 has been shortened.

Line 316: Accepted. We add suggested references and remove details about the NAO.

Lines 351: Acknowledged. In the revised manuscript we l remedy the problem

Line 393: Agreed. Modified.

Line 464: Accepted.  We will specify the IOPs.

Line 605: Accepted. The squares show the precipitation. We prepare Figure 11, now Figure 14, where the squares are distinguishes from the shaded background.

Line 640: We agree. The information about the data used in the data assimilation has been added.

[revised manuscript text omitted]

Line 672:  The verification measures (Table 3) show that the

Line 673: simulation with data assimilation produced slightly better results.  The scores for the entirety

Line 674: of Croatia show that the strong precipitation category results

Line 675: were improved for the operational run (CSI=0.28) compared to the run without data assimilation

Line 676: (CSI=0.23).  In addition, PCC showed that the  model and

Line 677: observations for the run with data assimilation were better associated.  The impact of data

Line 678: assimilation for that IOP  is rather small, but it yielded an improvement in the 24-

Line 679: hour precipitation forecast. It should be considered that for the selected case,

Line 680: better results were obtained with the higher resolution model and that the data assimilated in the

Line 681: operational ALADIN 8 km model  was mainly synoptic data. Thus, implementing data

Line 682: assimilation in the higher resolution model and adding additional high-resolution

Line 683: temporal and/or spatial data to the data assimilation system are apparently good ways to

[revised manuscript text omitted]